# The structure of behavioral variation within a genotype

Zachary Werkhoven[1], Alyssa Bravin[1], Kyobi Skutt-Kakaria[1,2], Pablo Reimers[1,3], Luisa F Pallares[4], Julien Ayroles[1,4], Benjamin L de Bivort[1]*

[1]Center for Brain Science and Department of Organismic and Evolutionary Biology, Cambridge, United States; [2]Division of Biology and Biological Engineering, California Institute of Technology, Pasadena, United States; [3]Janelia Research Campus, Howard Hughes Medical Institute, Ashburn, United States; [4]Department of Ecology and Evolutionary Biology and Lewis-Sigler Institute for Integrative Genomics, Princeton University, Princeton, United States

**Abstract** Individual animals vary in their behaviors. This is true even when they share the same genotype and were reared in the same environment. Clusters of covarying behaviors constitute behavioral syndromes, and an individual's position along such axes of covariation is a representation of their personality. Despite these conceptual frameworks, the structure of behavioral covariation within a genotype is essentially uncharacterized and its mechanistic origins unknown. Passing hundreds of inbred *Drosophila* individuals through an experimental pipeline that captured hundreds of behavioral measures, we found sparse but significant correlations among small sets of behaviors. Thus, the space of behavioral variation has many independent dimensions. Manipulating the physiology of the brain, and specific neural populations, altered specific correlations. We also observed that variation in gene expression can predict an individual's position on some behavioral axes. This work represents the first steps in understanding the biological mechanisms determining the structure of behavioral variation within a genotype.

*For correspondence: debivort@oeb.harvard.edu

Competing interest: The authors declare that no competing interests exist.

## Introduction

Individuals display idiosyncratic differences in behavior that often persist through time and are robust to situational context. Some persistent individual behavioral traits commonly occur in correlated groups and can therefore be said to covary. Behavioral ethologists have long understood that types of human and animal personalities often fall on multivariate axes of variation. A five-dimensional model (*Goldberg, 1993*) known as The Big Five personality traits is frequently used by psychologists to describe the range of human personality, and a similar model has been used to describe personality in fish (*Réale et al., 2007*). For example, human propensity for behaviors such as assertiveness, talkativeness, and impulsiveness is collectively described as extraversion and is thought to be anticorrelated with behaviors such as passivity, shyness, and deliberateness, all behaviors associated with introversion (*Matthews et al., 2003*).

In animal species, correlated suites of behaviors are described as behavioral syndromes. Aggressive behaviors such as fighting over mates or food are frequently correlated with exploratory behaviors such as foraging and social interaction. Correlated aggressive and exploratory behaviors have been observed in insects (*Jeanson and Weidenmüller, 2014*), arachnids (*Johnson and Sih, 2007*), fish (*Huntingford, 1976*), and birds (*van Oers et al., 2004*). The prevalence of behavioral correlation in so many species suggests that covariation is likely a universal feature of behavior. Although correlated individual differences in behavior are commonly observed, the structure and mechanisms of behavioral covariation are not well understood. Typically, where an individual lands on these behavioral axes

is thought to be established by a deterministic confluence of genetic and environmental effects. But there is increasing evidence that substantial individual behavioral variation is rooted in intragenotypic variation (*Honegger and de Bivort, 2018*; *Akhund-Zade et al., 2019*). The extent to which intragenotypic behavioral variation is organized into syndromes or axes is essentially uncharacterized.

Substantial variation in specific behavioral measures, even in inbred lines raised in standardized conditions, has been observed in several clonal animals, including geckoes (*Sakai, 2018*), amazonian mollies (*Bierbach et al., 2017*), and aphids and nematodes (*Schuett et al., 2011*; *Stern et al., 2017*). Genetic model systems hold particular promise for the mechanistic dissection of this variation, and intragenotypic variability (IGV) in behavior has been characterized in mice (*Freund et al., 2013*), zebrafish (*Bierbach et al., 2017*), and *Drosophila*. In flies, IGV of many behaviors has been studied, including phototaxis (*Kain et al., 2012*), locomotor handedness and wing-folding (*Buchanan et al., 2015*), spontaneous microbehaviors (*Kain et al., 2013*; *Todd et al., 2017*), thermal preference (*Kain et al., 2015*), and object-fixated locomotion (*Linneweber et al., 2020*). Mechanistic studies of these behavioral phenomena have addressed two major questions: (1) what biological mechanisms underlie the magnitude of behavioral variability (e.g., genetic variation [*Ayroles et al., 2015*] or neural state variation [*Kain et al., 2012*; *Buchanan et al., 2015*]), and (2) what specific differences within individual nervous systems predict individual behavioral biases (*Linneweber et al., 2020*; *Mellert et al., 2016*). The mechanistic basis of individuality is an exciting new field, but no study to date has focused on characterizing the large-scale variance-covariance structure of IGV in behavior.

Behavioral correlations within a genotype could arise through a number of biological mechanisms including cell-to-cell variation in gene expression or individual differences in neural circuit wiring or synaptic weights. For example, stochastic variation during developmental critical windows has the potential to impart lasting differences between individuals in the absence of conspicuous genetic or environmental differences. Any such differences affecting nodes common to multiple behaviors in neural or molecular pathways may result in correlated shifts in behavior. Here, we study this directly by focusing on the correlation structure of behavioral variation when genetic and environmental variation are minimized.

This is an important biological question for several reasons. The structure of intragenotypic behavioral variability (1) will shape the distribution and kinds of personalities that a population of organisms displays, even when they have matched genomes and environments, (2) is the product of stochastic biological outcomes, and its organization reveals how stochasticity drives variation, (3) constrains the evolution of behavior and adaptive phenotypic strategies like bet-hedging (*Hopper, 1999*), (4) is a relatively uncharacterized component of neural diversity and its manifestations in behavior and disease, and (5) may shed light on how the nervous system orchestrates behavior as a whole. In flies, we have a suitable experimental system for directly characterizing this structure as we can produce large numbers of individuals with nearly identical genomes, reared in the same environment, and collect many behavioral measures per individual. We performed this experiment in wild-type inbred flies as well as wild-type outbred flies and collections of transgenic lines manipulating neural activity. This approach let us contrast the structure of intragenotypic behavioral variability in animals where the source of variability is, respectively, stochastic fluctuations, genetic differences + stochastic fluctuations, and systematic perturbations of the nervous system + stochastic fluctuations. We found that in all cases, behavioral variation has high dimensionality, that is, many independent axes of variation. The addition of variation from genetic differences and neural perturbations did not fundamentally alter this qualitative result, suggesting that stochastic fluctuations and genetic differences may structure behavior through common biological mechanisms.

## Results

### High-dimensional measurement of individual behaviors

The first step in revealing the structure of behavioral variation within a genotype was to devise an experimental pipeline that produced a data set of many (200+) individual flies, with many behavioral measurements each. We developed a number of behavioral assays, measuring both spontaneous and stimulus-evoked responses of individual flies, which could be implemented in a common experimental platform (*Figure 1A*; *Werkhoven et al., 2019*). This instrument features an imaging plane, within which flies moved in arenas of various geometries. Fly position was tracked with digital cameras

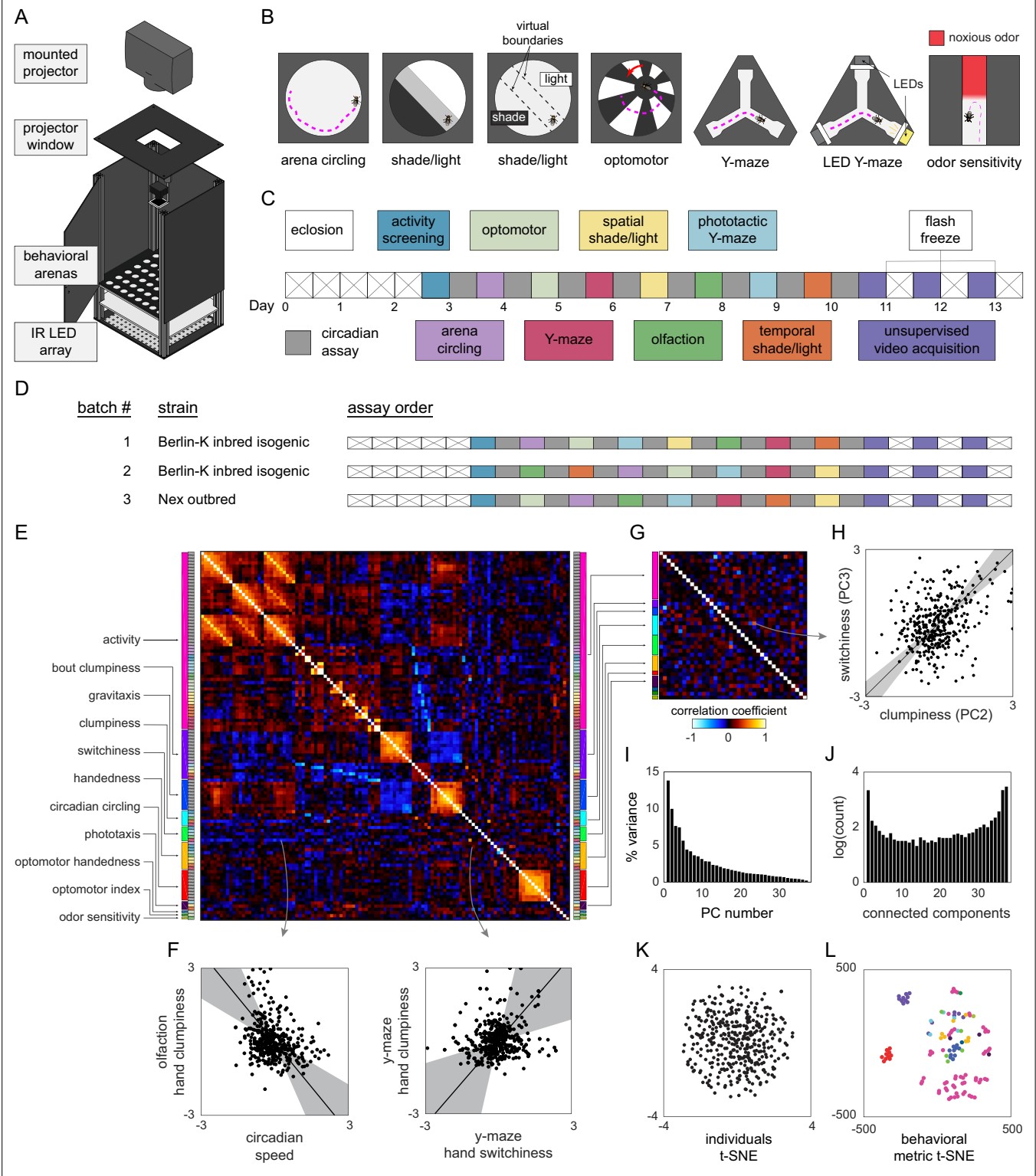

**Figure 1.** Decathlon experimental design and structure of intragenotypic behavioral variation. (**A**) Schematic of the imaging rig used for most Decathlon experiments. (**B**) Schematics of the behavioral assays, illustrating the geometry of the arenas and stimulus structure. (**C**) Timeline of the Decathlon experiment. Colors indicate the assays conducted on each day, half-black half-white blocks indicate the circadian assay and storage in 96-well plates. (**D**) Timelines of the three Decathlon experiments, indicating the randomized order of assays 2–8. (**E**) Full correlation matrix of all raw behavioral measures taken in the Decathlon. Colored blocks indicate blocks of measures we thought a priori might be correlated (outer blocks, text labels). Inner blocks indicate assay. (**F**) Example scatter plots associated with measure correlations. Points are individual flies. Line is the best fit (principal component [PC]1

*Figure 1 continued on next page*

*Figure 1 continued*

of these points), gray region is the 95% confidence interval of the fit, as determined by bootstrap resampling. (**G**) Distilled correlation matrix in which all correlated measures represent unexpected relationships. Meaningful correlations in this matrix can be found outside the within-a-priori-group on-diagonal blocks. (**H**) Example scatter plot from the distilled correlations. Plot elements as in (**F**). (**I**) Scree plot of the ranked, normalized eigenvalues, that is, the % variance explained by each PC, of the distilled behavior matrix, versus PC #. (**J**) Connected components spectrum (see text and *Figure 1—figure supplement 10*) for the distilled correlation matrix. Height of bars indicates organization at that dimensionality. (**K**) Points corresponding to individual flies nonlinearly embedded using *t*-SNE from the 121-dimensional full matrix to two dimensions. (**L**) Points corresponding to behavioral measures nonlinearly embedded using *t*-SNE from the 384-dimensional space of flies to two dimensions. Colors indicate groups of measures we expected a priori to be related.

The online version of this article includes the following figure supplement(s) for figure 1:

**Figure supplement 1.** Persistence of primary behavioral measures across assays.

**Figure supplement 2.** Genomic sequencing to confirm isogeny of Berlin-K$^{iso}$.

**Figure supplement 3.** Decathlon behavioral measure distributions.

**Figure supplement 4.** Schematic of the Decathlon analysis pipeline.

**Figure supplement 5.** Simulated comparison of matrix-infilling methods.

**Figure supplement 6.** Correlation of correlation matrix values between inbred-1 vs. inbred-2 and shuffled matrices.

**Figure supplement 7.** Principal component analysis (PCA) of a priori groups.

**Figure supplement 8.** Distribution of correlation coefficients and p-values in the full and distilled correlation matrix.

**Figure supplement 9.** Measure loadings for the handedness and activity a priori groups.

**Figure supplement 10.** Measure loadings for various a priori groups.

**Figure supplement 11.** Correlations among the principal components (PCs) of switchiness and clumpiness.

**Figure supplement 12.** Connected components spectra for determining dimensional organization: simulated data examples.

using diffused infrared illumination invisible to the flies. Visual stimuli were presented to the animals using DLP projectors or LEDs embedded in the arena walls. We implemented six assays (*Supplementary file 1*) in this style, assessing (1) spontaneous walking in circular arenas, (2) preference to rest in brighter or dimmer positions (in an environment of spatially structured illumination), (3) preference to rest in brighter or dimmer light levels (in a fictive, temporally modulated light environment), (4) optomotor responses to rotating visual stripes, (5) spontaneous left-right decision making in Y-mazes, and (6) phototaxis in Y-mazes, where flies are given a choice of walking toward or away from a lit LED (*Figure 1B*).

To these assays, we added three more, assessing (7) odor sensitivity in linear chambers (*Claridge-Chang et al., 2009*; *Honegger et al., 2020*) in which half of the compartment is filled with an aversive odorant, (8) spontaneous behavior, acquired via high-resolution 100 Hz video and suitable for pixel-based unsupervised classification (*Berman et al., 2014*), and (9) circadian activity and spontaneous locomotion in 96-well plates with access to food. Each of these assays produced multiple behavioral measures for each individual fly. For example, flies behaving in the phototactic 'LED Y-maze' (assay 6) are performing phototaxis and exploratory locomotion but yield several different behavioral measures, including the number of choices made by passing through the choice point of the Y-maze (a measure of total activity), the fraction of turns that are to the right, the fraction of turns that are toward the lit LED, the number of pauses in which the animal did not move, the average duration of pauses, etc. Thus, the total collection of behavioral measures across all assays per fly was quite large (up to 121), constituting a diverse, inclusive characterization of individual behavior. Each assay has a particular measure that captures the behavior it is primarily designed to assess (e.g., the fraction of turns toward the lit LED in the LED Y-maze). In control experiments, we confirmed that these primary measures are consistent across days within an individual (*Figure 1—figure supplement 1*; i.e., they reflect persistent idiosyncrasies [*Kain et al., 2012*; *Buchanan et al., 2015*; *Kain et al., 2015*; *Honegger et al., 2020*]).

In order to obtain all of the behavioral measures from each experimental animal, we combined our behavioral assays in a serial experimental pipeline lasting 13 consecutive days (*Figure 1C*), generally with one unique assay per day and continuous circadian imaging (assay 9) between assays. This pipeline begins with 3-day-old flies being loaded into the 96-well circadian imaging plates. Using a common behavioral platform as much as possible and storing flies between experiments in 96-well

plates made maintaining the errorless identity of flies over the whole 13 -day experiment substantially easier. Starting on day 3, daily assays began. On each day, flies were lightly anesthetized on an ice-chilled plate and aspirated, maintaining their identity, into the assay arrays. After the assay was completed (typically after 2 hr of recording), flies were again lightly anesthetized and returned to 96-well plates for renewed circadian imaging. On the first such day, flies were loaded into an array of circular arenas and imaged for total activity (in a version of assay 1). At this point, the most active 192 flies were retained for further testing. In preliminary experiments, we found that flies that were inactive at the beginning of the pipeline were very unlikely to produce substantial amounts of data over the rest of the pipeline. With the addition of this activity-screening assay, the total number of experiments was 10, and as each fly 'competes' in all 10 events, we refer to the entire pipeline as a Decathlon.

It is possible that the assay order has some effect on the recorded behavioral measures. So we randomized the assay order between Decathlon implementations as much as possible (*Figure 1D*), subject to two restrictions: activity screening was always the first assay, and high-resolution imaging for unsupervised analysis (assay 8) was always the last assay. (This assay has lower throughput, and 3 days were required to complete all 168 remaining flies. If this assay were performed earlier in the pipeline, it might introduce heterogeneity across subsequent assays.) When each fly completed its run through all Decathlon assays (i.e., over the 3 days of assay 8 imaging), it was flash-frozen in liquid nitrogen for RNA sequencing.

## Behavioral variability among inbred flies has high dimensionality and sparse pairwise correlations

To collect data that would reveal the structure of behavioral variation within a genotype, we conducted two Decathlons using highly inbred, nearly isogenic flies derived from the wild-type strain Berlin-K (BSC#8522; *Nöthel, 1981*). We confirmed that this strain was, indeed, highly isogenic with genomic sequencing of individual animals, finding ~75 SNPs in the population across the entire genome (*Figure 1—figure supplement 2*). 115 flies completed the first Decathlon, and 176 the second. While we aimed to collect 121 measures per fly, a portion of values were missing, typically because flies did not meet assay-specific activity cutoffs. The sample sizes achieved for each assay and the distributions of all raw measures are given in *Figure 1—figure supplement 3*. For subsequent analyses, it was sometimes necessary to have a complete data matrix (see *Figure 1—figure supplement 4* for a schematic of all analysis pipelines). We infilled missing values using the alternating least-squares method, which, as judged by analyses of toy ground truth data, performed better than mean-infilling (*Figure 1—figure supplement 5*). For the sake of maximal statistical power, we wanted to merge the data sets from the two Berlin-K^iso Decathlons. The correlation matrices of these two data sets were not identical, but were substantially more similar than expected by chance (*Figure 1—figure supplement 6*), implying that while there were inter-Decathlon effects, much of the same structure was present in each and merging them was justified. To do this, we z-score-normalized the data points from each arena array/batch (within each Decathlon) across flies, thus eliminating any arena, assay, and Decathlon effects and enriching the data for contrasts between individuals. A grand data matrix was made by concatenating these batches (382 individuals × 121 behavioral measures).

The full correlation matrix of this Berlin-K^iso data set is shown in *Figure 1E*. It contains a substantial amount of structure, indicating that large groups of behavioral measures covary. But the covariance of many pairs of measures in the matrix is not surprising. For example, almost all our assays generate some measure of locomotor activity (meanSpeed in circular arenas, number of turns in Y-mazes, meanSpeed in the olfactory tunnels, etc.), and one might expect that especially active flies in one assay will be especially active in another assay. Additional unsurprising structure in this matrix comes in identical measures recorded in each of the 11–13 circadian assays each fly completed. But, even in this first analysis, surprising correlations were evident. For example, flies with higher variation in the inter-turn interval in the olfactory assay ('clumpiness' in their olfactory turning) exhibited higher mean speed in the circadian assays, and flies with higher variation in inter-turn intervals in the Y-maze (clumpiness in their Y-maze turning) exhibited lower mutual information in the direction of subsequent turns in the Y-maze ('switchiness' in their Y-maze handedness; *Figure 1F*. See below and *Buchanan et al., 2015*; *Akhund-Zade et al., 2019*) for more about these measures.

To produce an exhaustive list of such non-trivial correlations, we distilled the grand correlation matrix to a smaller matrix (the 'distilled matrix'; *Figure 1G*) in which two kinds of interesting relationships were revealed: (1) uncorrelated dimensions among measures for which we had a prior expectation of correlation (e.g., if meanSpeed in circular arenas is found to be uncorrelated with meanSpeed in olfactory tunnels), and (2) correlated dimensions among measures for which we had no prior expectation of correlation. Relationships of the former class were identified by enumerating, before we ran any correlation analyses, groups of measures we expected to be correlated ('a priori groups'; *Figure 1E and G*). See *Supplementary file 2* and *Supplementary file 3* for a breakdown of the measures included in each a priori group and their respective inclusion criteria. We looked for surprising independence within such groups by computing the principal components (PCs) of data submatrices defined by the grouping (e.g., for the 'activity' a priori group, by running principal component analysis (PCA) on the data set consisting of 382 individual flies, and 57 nominal measures of activity). We then replaced each a priori group submatrix with its projection onto its statistically significant PCs, as determined by a reshuffling analysis (see Materials and methods, *Figure 1—figure supplement 7*). Some a priori groups largely matched our expectations, with relatively few uncorrelated dimensions among many measures (e.g., the gravitaxis group that had 10 measures and only 2 significant PCs), while others exhibited relatively many uncorrelated dimensions (e.g., the clumpiness group that had five measures and five significant PCs; see *Figure 1—figure supplement 7* for all a priori group PCAs).

With a priori group submatrices represented in their respective significant PCs, the grand data matrix now contained 38 behavioral measures. Every significant correlation between behavioral measures at this point represents an unexpected element of structure of behavioral variation (*Figure 1G*). Given the high dimensionality of the full data set and the complex structure of the behavioral measures in the distilled correlation matrix, we created an online data browser (http://decathlon.debivort.org) to explore the data and compare the alternative correlation matrices. The first impression of the distilled correlation matrix is that it is sparse. Most behavioral measures are uncorrelated or weakly correlated, suggesting that there are many independent dimensions of behavioral variation. However, 176 pairs of behaviors were significantly correlated at a false discovery rate (FDR) of 38%, and the distribution of p-values for the entries in this matrix exhibits a clear enrichment of low values (*Figure 1—figure supplement 8*), indicating an enrichment of significant correlations. As an example, flies with high values in the first PC of the phototaxis a priori group tend to have high values in the second PC of the activity level a priori group. As another example, the third switchiness PC is positively correlated with the second clumpiness PC (*Figure 1H*; a relationship that is likely related to the positive correlation between the Y-maze hand clumpiness and Y-maze hand switchiness, *Figure 1F*). Interpreting the loadings (*Figure 1—figure supplements 9 and 10*) of these PCs indicates that this is a correlation between olfactory tunnel turn direction switchiness and olfactory tunnel turn timing clumpiness. We detected a substantial number of correlations between different dimensions of switchiness and clumpiness (*Figure 1—figure supplement 11*), suggesting that there are multiple couplings between these suites of traits.

Stepping back from specific pairwise correlations, we examined the overall geometry of behavioral variation. The full matrix contained 22 significant PCs, with PCs 1–3 explaining 9.3, 6.9, and 5.8% of the variance, respectively (the distilled matrix contained 16 significant PCs, with PCs 1–3 explaining 13.8, 10.0, and 7.7% of the variance, respectively; *Figure 1I*). But the amount of variance explained across PCs does not provide the full picture of how many dimensions of variation are present in a data set. A correlation matrix can be organized at different scales/hierarchically, so there need not be a single number that characterizes dimensionality. As an intuitive example, data filling a volume shaped like a frisbee is organized largely in two dimensions. Data shaped like a rugby ball is somewhat one-dimensional, not particularly two-dimensional, and somewhat three-dimensional. An approach is needed that can characterize such continuous variation in organization across dimensionalities, particularly the possibility that the data are not structured tidily in a single dimensionality.

We developed a 'connected components spectrum' analysis that characterizes the continuously varying degree of organization of a correlation matrix from dimensionality 1 to $d$, the total dimensionality of the data. Briefly, we thresholded the absolute value of the correlation matrix at values ranging from 1 to $d$, and, treating these matrices as adjacency matrices, determined the number of connected components, recording how often $n$ connected components were observed. See Materials

and methods and *Figure 1—figure supplement 12*. These connected components spectra can be interpreted as follows: peaks at dimensionality = 1 indicate that all measures are coupled in a network of at least weak correlations; peaks at dimensionality = *d* indicate that all measures are to some degree uncorrelated; peaks in between these values indicate intermediate scales of organization. Multiple peaks are possible because these kinds of organization are not mutually exclusive. The connected components spectrum (*Figure 1J*) of the distilled Decathlon data set had peaks at 1 and *d* (*Li and Durbin, 2009*). There was also evidence for structure over the full range of intermediate dimensionalities. Overall, the organization is one of predominantly uncorrelated behaviors, with sparse sets of behaviors correlated with continuously varying strengths.

To assess how individual flies are distributed in behavior space, we embedded them from the 384-dimensional space into two dimensions using *t*-SNE (*van der Maaten and Hinton, 2008*). There appear to be no discrete clusters corresponding to 'types' of flies. Instead, variation among flies appears continuously distributed around a single mode (*Figure 1K*). The same is true for an embedding of flies as represented in the distilled matrix (*Figure 4—figure supplement 3D*). We also embedded behavioral measures as points from the 121-dimensional space of flies into two dimensions (*Figure 1L*). This confirmed that while our intuition for which sets of measures would be similar (the a priori groups) was right in many cases, measures we thought would be similar were often dissimilar across flies, and sometimes measures we did not anticipate being similar were (e.g., phototaxis and activity level).

## Behaviors that covary among individuals tend to be patterned similarly in time and across the body

With data from the second Decathlon, we characterized the structure of variation in a set of behaviors that was potentially exhaustive for one behavioral condition (free walking/motion in a 2D arena; *Figure 2A*). High-speed, high-resolution video was acquired for flies simultaneously in each of two rigs. Over 3 days, we acquired 13.5 GB of 200 × 200 px 100 Hz videos centered on each fly as they behaved spontaneously over the course of 60 min. These frames were fed into an unsupervised analysis pipeline (*Berman et al., 2014*) that computed high-dimensional representations of these data in the time-frequency domain before embedding them in two dimensions and demarcating boundaries between 70 discrete modes of behavior (*Figure 2B*). The behavior of each fly was thus represented as one of 70 values at each frame. Flies exhibited a broadly similar probability distribution of performing each of these behaviors (*Figure 2C*), though there were conspicuous differences among individual patterns of behavior (*Figure 2D*).

The correlation matrix of behavioral modes identified in the unsupervised analysis was highly structured (*Figure 2E*; like the correlation matrix of the rest of the Decathlon measures, *Figure 1E*), appearing to be strongly organized in 10 or fewer dimensions, with some evidence of organization in higher dimensions (*Figure 2F*). Correlations between behavioral modes identified by unsupervised clustering and measures from the other Decathlon assays were generally weaker than correlations within these categories, but were nevertheless enriched for significant relationships (*Figure 2—figure supplement 1*). For this analysis, there was no equivalent of a priori groups of behavioral measures as measures were not defined prior to the analysis. But, in examining sample movies of flies executing each of the 70 unsupervised behavioral modes (*Videos 1–4*), it was clear that highly correlated behavioral modes tended to reflect variations on the same type of behavior (e.g., walking) or behaviors performed on the same region of the body (e.g., anterior movements including eye and foreleg grooming; *Figure 2G*). In other words, individual flies that perform more eye grooming tend to perform more of other anterior behaviors. There were some correlations between behaviors implemented by disparate parts of the body. For example, flies that spent more time performing anterior grooming also spent more time performing slow leg movements (*Figure 2G*). The overall similarity of covarying behaviors was confirmed by defining groups of covarying behaviors and observing that they were associated with contiguous regions of the embedded behavioral map (*Figure 2G*). That is, behaviors whose prevalence covaries across individuals have similar time-frequency patterns across the body. Moreover, these clusters of covarying, contiguously embedded behaviors exhibited similar temporal transitions; behaviors that covary across individuals tend to precede specific sets of subsequent behaviors (*Figure 2H*). Thus, there appear to be couplings between the dimensions of behavioral variation across individuals, the domains of the body implementing behavior, and the temporal patterning of behaviors.

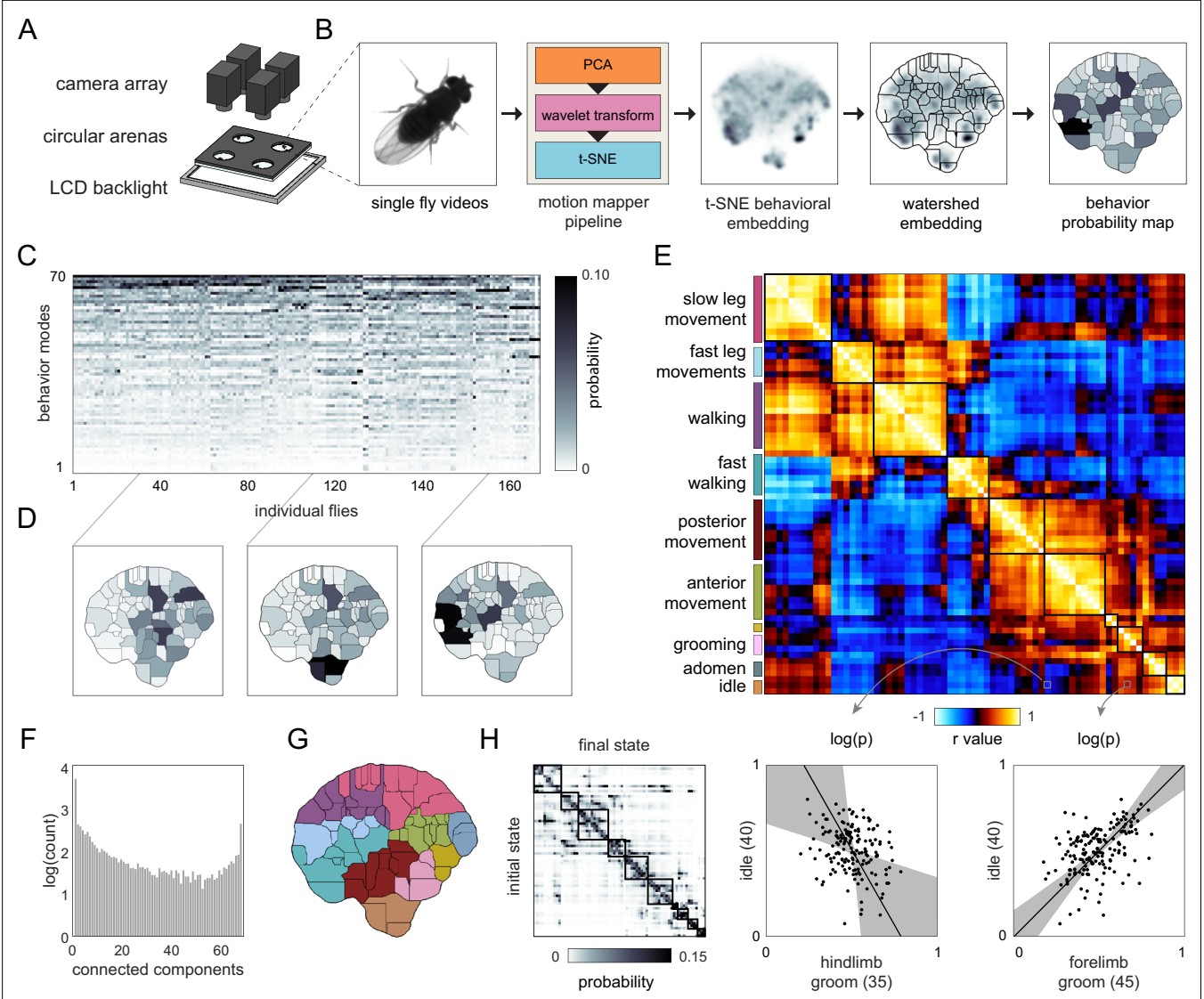

**Figure 2.** Correlation structure of unsupervised behavioral classifications. (**A**) Schematic of the four camera imaging rig used to acquire single fly videos. (**B**) Overview of the data processing pipeline from single fly videos to behavioral probability maps. (**C**) Sample individual behavior mode probability density functions (PDFs visualized in an embedded *t*-SNE space). Discrete regions correspond to watersheds of the *t*-SNE embedding. (**D**) Sample individual PDFs mapped to locations in *t*-SNE space. Discrete regions correspond to watersheds of the *t*-SNE embedded probability densities. (**E**) Correlation matrix (top) for individual PDFs with rows and columns hierarchically clustered. Colored blocks indicate labels applied to classifications post hoc. Example scatter plots (bottom) of individual behavioral probabilities. Points correspond to probabilities for individual flies. Line is the best fit (principal component [PC]1 of these points), gray region is the 95% confidence interval of the fit, as determined by bootstrap resampling. (**F**) Connected components histogram of the thresholded PDF correlation matrix (see Materials and methods). (**G**) Discrete behavioral map with individuals zones colored by post hoc labels as in (**E**). (**H**) Transition probability matrix for behavioral classifications. Entries in the *i*th row and *j*th column correspond to the probability of transitioning from state *i* to state *j* over consecutive frames. Blocks on the diagonal indicate clusters of post hoc labels as in (**E**) and (**G**).

The online version of this article includes the following figure supplement(s) for figure 2:

**Figure supplement 1.** Correlation of Decathlon assay and unsupervised behavioral measures.

## Thermogenetic neural perturbation alters the correlations between behaviors

To (1) confirm that the Decathlon experiments revealed biologically meaningful couplings between behaviors and (2) probe biological mechanisms potentially giving rise to behavioral correlations, we treated correlations in the Decathlon matrices as hypotheses to test using data from a thermogenetic neural circuit perturbation screen (*Skutt-Kakaria et al., 2019*). Specifically, we focused on the

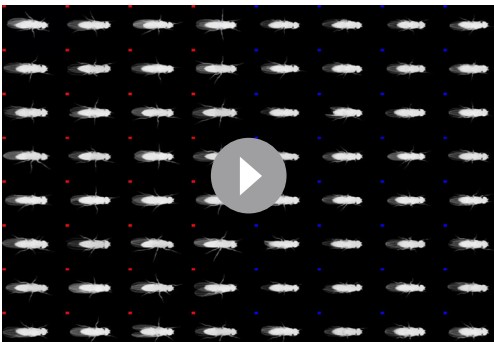

**Video 1.** Examples of a mode of walking behavior as identified by the unsupervised analysis, from movies of single flies, made up of successive frames classified as the same behavior. Colored dots indicate whether flies are outbred (NEX; red) or inbred (Berlin-K^iso; blue).
https://elifesciences.org/articles/64988/figures#video1

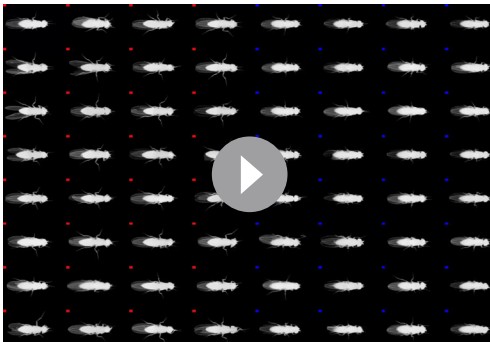

**Video 3.** Examples of a mode of head grooming behavior as identified by the unsupervised analysis, from movies of single flies, made up of successive frames classified as the same behavior. Colored dots indicate whether flies are outbred (NEX; red) or inbred (Berlin-K^iso; blue).
https://elifesciences.org/articles/64988/figures#video3

many correlations between measures of turn timing clumpiness and turn direction switchiness (*Figure 1H* and S8). Before the Decathlon experiment, we had no reason to think these measures would be correlated, as one describes higher-order structure in the timing of locomotor turns (clumpiness), and the other describes higher-order structure in the direction of sequential turns (switchiness). Our prediction derived from the Decathlon was that if perturbing a circuit element caused a change in clumpiness it would tend to also cause a change in switchiness in a consistent direction. We looked for such correlated changes when we inactivated or activated neurons in the central complex (*Figure 3—figure supplement 1A–C*), a cluster of neuropils involved in locomotor behaviors (*Buchanan et al., 2015*; *Ofstad et al., 2011*; *Kottler et al., 2019*) and heading estimation (*Seelig and Jayaraman, 2015*; *Green et al., 2017*; *Kakaria and de Bivort, 2017*). We used a set of Gal4 lines (*Wolff and Rubin, 2018*), each of which targets a single cell type and that tile the entire protocerebral bridge (a central complex neuropil; see *Supplementary file 4*), to express Shibire^ts (*Kitamoto, 2001*) or dTRPA1 (*Hamada et al., 2008*), thermogenetic reagents that block vesicular release and depolarize cells, respectively. As controls, we used flies heterozygous for the Gal4 lines and lacking the effector transgenes.

Flies with these genotypes were loaded into Y-mazes for behavior imaging before, during, and after a temperature ramp from the permissive temperature (23 °C) to the restrictive temperature (32 °C for

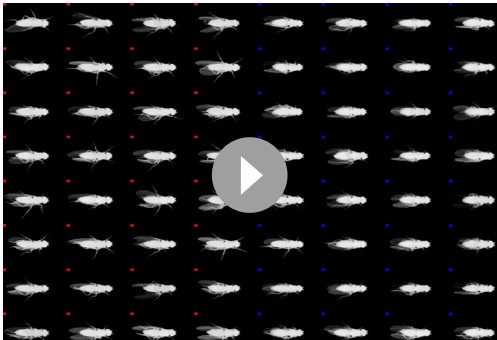

**Video 2.** Examples of a mode of wing grooming behavior as identified by the unsupervised analysis, from movies of single flies, made up of successive frames classified as the same behavior. Colored dots indicate whether flies are outbred (NEX; red) or inbred (Berlin-K^iso; blue).
https://elifesciences.org/articles/64988/figures#video2

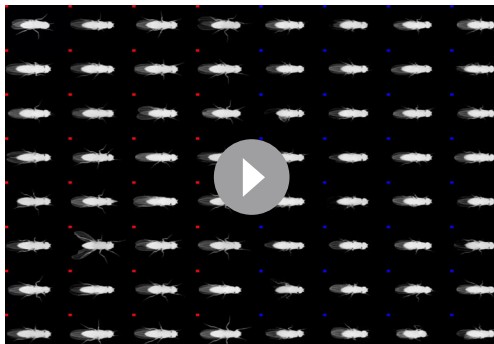

**Video 4.** Examples of a mode of abdomen flexing behavior as identified by the unsupervised analysis, from movies of single flies, made up of successive frames classified as the same behavior. Colored dots indicate whether flies are outbred (NEX; red) or inbred (Berlin-K^iso; blue).
https://elifesciences.org/articles/64988/figures#video4

dTRPA1 and 29 °C for Shibire[ts]) (*Figure 3—figure supplement 1A*). At the permissive temperature, we observed significant negative correlations in the average line clumpiness and switchiness of control and dTrpA1 expressing lines (*Figure 3—figure supplement 1D and E*). This suggests that the mechanisms that couple variation in switchiness and clumpiness within a genotype may also be at play across genotypes. Surprisingly, at the restrictive temperature, we saw significant positive correlations between clumpiness and switchiness in all three experimental treatments: control (Gal4/+), Gal4/Shibire[ts], and Gal4/dTRPA1 lines. That this correlation appeared in controls suggests that temperature alone can selectively alter the function of circuit elements regulating both clumpiness and switchiness, effectively reversing their coupling. The dTrpA1- and Shibire[ts]-expressing lines also showed this reversal, but to a lesser extent, suggesting that perturbing neurons in the central complex can block temperature-induced changes in the coupling of clumpiness and switchiness.

## Individual gene expression variation correlates with individual behaviors

That thermogenetic manipulation can disrupt correlations between behavioral measures suggests that specific patterns of neural activity underlie the structure of behavioral variation. Such physiological variation could arise in stochastic variation in gene expression (*Lin et al., 2016*) in circuit elements. To test this hypothesis, we performed RNA sequencing on the heads of the flies at the end of the first Decathlon experiment (*Figure 3A*). We used Tm3'seq (*Pallares et al., 2020*) to make 3'-biased libraries for each individual animal. We quantified the expression of 17,470 genes in 101 individual flies. The expression profiles were strongly correlated across individuals (*Figure 3B*), but there was some significant variation across individuals (*Figure 3C*). To assess whether this variation was meaningful with respect to behavioral variation, we trained linear models (over 625,000 in total) to predict an individual's behavioral measures from its transcriptional idiosyncrasies. Specifically, we fit simple linear models for each of our 97 behavioral measures as a function of the 6642 most highly expressed genes. The median model had an $r^2$ value of 0.008% and 5% of models predicted behavior at $r^2 >$ 0.135. Behavioral measures varied greatly in their number of significant ($p<0.05$) gene predictors (*Figure 3D*), ranging from 147 to 1172 genes.

After identifying genes that were predictive of each behavioral measure, we assessed whether they shared functional characteristics (*Figure 4—figure supplement 1A*) using KEGG pathway enrichment analysis (*Mootha et al., 2003*; *Kanehisa and Goto, 2000*; *Yu et al., 2012*). We independently performed KEGG enrichment analysis on each gene list to identify functional categories of genes overrepresented for any particular behavioral measure compared to the background list of 6642 genes. Of this background, 1982 genes (30%) were associated with one or more KEGG pathways. Across all behaviors, we found that 37 KEGG pathways were significantly enriched in the genes that predicted individual behavioral measures (*Figure 3E*, *Figure 4—figure supplement 1*, *Supplementary file 5*). Many of these pathways were significantly enriched compared to shuffled controls and were enriched in multiple behavioral measures (*Figure 3F*). We included features in the online Decathlon Data Browser to explore gene-behavior correlations (searching either by gene or by behavior) as well as KEGG pathway-behavior correlations. We repeated this experiment and analysis on flies of the second Decathlon experiment and found that the significance of pathway enrichments was highly correlated ($r = 0.87$; *Figure 3G*), with 43 and 73% of significant KEGG pathways in the first and second Decathlon experiments (respectively) shared in the other experiment.

Genes related to cellular respiration, protein translation, and phototransduction were significantly enriched for multiple behavioral measures, a result that was highly robust to bootstrap resampling. Cellular respiration was the most common enrichment, significant in 11 of 97 behavioral measures, suggesting that variation in metabolic rate may be predictive of variation in many behaviors. Indeed, metabolic function was a common link between significant categories, with a total of 24 of 37 being related to metabolism. We also found 11 pathways (including two highly enriched categories: ribosome and proteasome) associated with protein turnover. While most enriched pathways were related to basic cellular processes, others (Hedgehog signaling, Wnt signaling, insect hormone biosynthesis, SNARE vesicular transport, and phototransduction) suggested roles for development and neuronal function in individual behavioral variation.

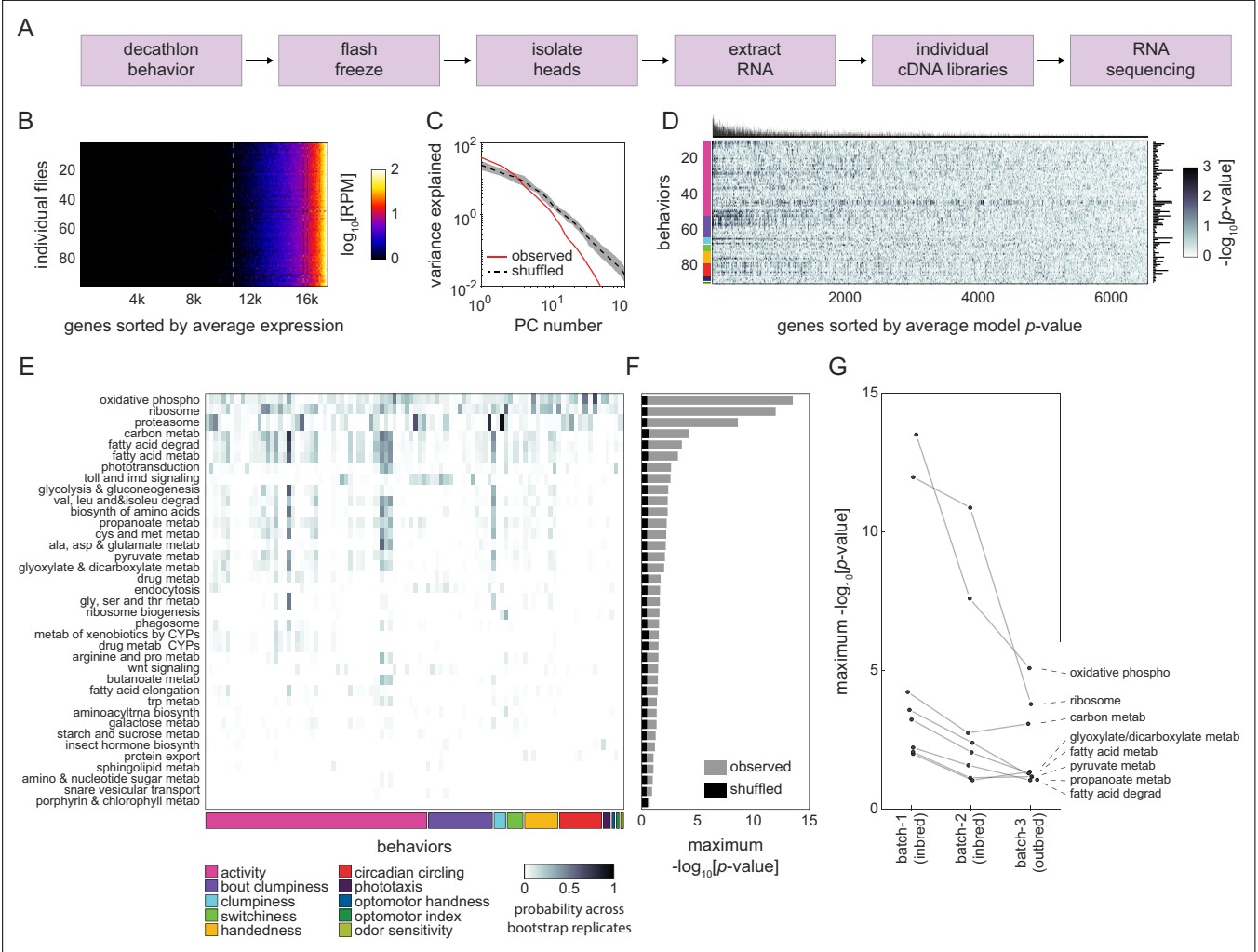

**Figure 3.** Correlation between individual transcriptomes and behavioral biases. (**A**) Steps for collecting transcriptomes from flies that have completed the Decathlon. (**B**) Data matrix of individual head transcriptomes. Rows are individual flies (n = 98). Columns are 17,470 genes sorted by their mean expression across individuals. The dashed line indicates the mean expression cutoff at 10 reads per million (RPM), below which genes were excluded from analysis. (**C**) Scree plots of the logged % variance explained for ranked principal components of gene expression variation, for observed (red) and shuffled (black) data. Shaded region corresponds to 95% CI as calculated by bootstrap resampling. (**D**) Performance heatmap (-log p) of linear models predicting the behavior of individual flies from single-gene expression. Colored bars (left) indicate a priori group identity of behavioral measures (rows). Bar graphs show the number of significant (p<0.05) models for each gene (top) and behavior (right). (**E**) Heatmap showing the probability across bootstrap replicates of a KEGG pathway being significantly enriched in the list of predictive genes for a given behavior. (**F**) Bar plot showing the average across bootstrap replicates of the maximum (across behaviors) negative log adjusted p-value of all enriched KEGG pathways. Color indicates results from observed (gray) or shuffled control (black) data. (**G**) Average maximum adjusted -log p-value for enriched KEGG pathways common to all Decathlon iterations. Pathway labels (right) are ordered by batch 3 (outbred) -log adjusted p-value.

The online version of this article includes the following figure supplement(s) for figure 3:

**Figure supplement 1.** Effect of thermogenetic neural perturbation on clumpiness and switchiness.

## Behavioral variability has a similar structure in inbred and outbred lines

If transcriptomic differences predict individual behavioral differences within a genotype, then the structure of behavioral variability might be very different in outbred populations, where transcriptomic differences are (presumably) much more substantial. We tested this by conducting a Decathlon experiment on outbred flies (n = 192) from a synthetic genetic mapping population (*Long et al., 2014*). These animals were from a high (~100)-generation intercross population ('NEX'; seeded in the first generation by eight kinds of F$_1$ heterozygotes produced by round-robin cross from eight inbred wild strains). A distilled correlation matrix of behavioral measures (*Figure 4A*) was produced by the same

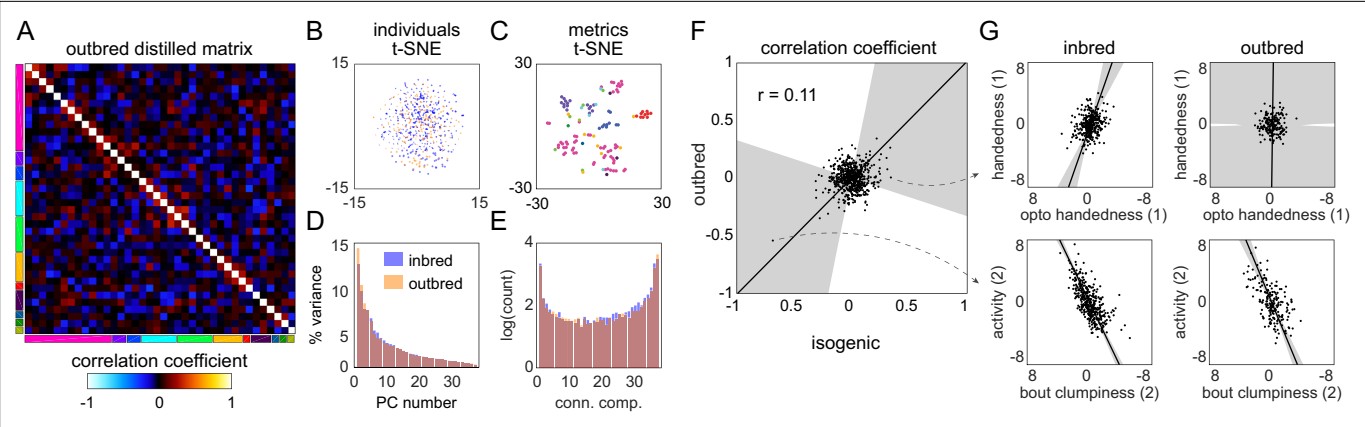

**Figure 4.** Structure of behavioral variation in outbred flies. (**A**) Distilled correlation matrix for outbred NEX flies. Colored blocks indicate a priori groups as described in *Figure 1*. (**B**) Points corresponding to individual flies nonlinearly embedded using *t*-SNE from the 121-dimensional full matrix to two dimensions. Color indicates whether the flies were inbred (blue) or outbred (orange). (**C**) Points corresponding to behavioral measures nonlinearly embedded using *t*-SNE from the 192-dimensional space of flies to two dimensions. Colors correspond to a priori group. (**D**) Scree plot of the ranked, normalized eigenvalues, that is, the % variance explained by each principal component (PC), of the distilled covariance matrix, versus PC #. (**E**) Connected components spectra for outbred and inbred correlation matrices (see Materials and methods). (**F**) Scatter plot of the distilled matrix correlation coefficients for inbred and outbred flies. Points correspond to distilled matrix measure pairs. (**G**) Example scatter plots of distilled matrix measure pairs for inbred (left) and outbred (right) flies. The rows of plots highlight a pair of measures with qualitatively different (top) and similar (bottom) correlations in inbred and outbred flies.

The online version of this article includes the following figure supplement(s) for figure 4:

**Figure supplement 1.** Correlation structure of outbred unsupervised behavior classifications.

**Figure supplement 2.** Correlation between individual transcriptomes and behavioral biases.

**Figure supplement 3.** Correlation between individual transcriptomes and unsupervised behavioral measures.

**Figure supplement 4.** Comparison of inbred and outbred correlations.

method as above. At first glance, it appears qualitatively similar to the distilled correlation matrix from inbred animals (*Figure 1G*). This impression was confirmed in more formal comparisons of the structure of behavior in inbred and outbred populations. In inbred and outbred populations, (1) individuals do not fall into discrete clusters, as determined by *t*-SNE embedding of individuals as points (*Figure 4B*). Moreover inbred and outbred flies appear to lie on the same manifold in behavioral measure space; (2) behavioral measures cluster according to their membership in a priori groups similarly in outbred (*Figure 4C*) and inbred (*Figure 1L*) populations; (3) the distribution of percent variance explained by PC was similar (*Figure 4D*); and (4) there is a similar connected components spectrum, with substantial distribution of behavioral biases across the full $d$ dimensions and a sparse network of correlations coupling behaviors over a continuum of dimensionalities (*Figure 4E*). In addition to the Decathlon assay measurements, we also recorded high-speed, high-resolution videos of outbred flies for unsupervised classification and co-embedded their postural behavior modes with those of inbred flies. Correlated clusters of unsupervised behavior modes from outbred flies corresponded broadly to the clusters mapped to anatomical regions in inbred flies (*Figure 4—figure supplement 1*). We further conducted transcriptomic sequencing and analysis on the heads of these outbred flies and found that, similar to the inbred flies, their behaviors were broadly correlated to gene expression (*Figure 4—figure supplement 2*, *Figure 4—figure supplement 3*). The percent of significant pathways overlapping between pairs of inbred and outbred Decathlon experiments was similar to that of pairs of inbred Decathlon experiments, ranging from 38% to 64%.

After determining that the overall structure of behavioral variation in inbred and outbred populations is similar, we asked whether it was also similar in specific correlations. There appears to be some similarity at this level (*Figure 4F*); the correlation coefficient between the inbred and outbred populations in the pairwise correlations between behavioral measures is statistically significant (p=0.001), but low in magnitude ($r = 0.11$). Examining specific pairs of scatter plots, it is clear that the correlations between specific behaviors are sometimes the same between the inbred and outbred animals, but sometimes not (*Figure 4G*, *Figure 4—figure supplement 4*). A caveat in interpreting apparent

differences between the inbred and outbred matrices is that two qualities are different between the animals used in the respective experiments: the degree of genetic diversity, but also (necessarily) the genetic background of the flies.

## Behavioral variability has high dimensionality regardless of the mechanistic origins of variation

Lastly, we examined how the correlation structure of behavior compared between sets of flies with variation coming from different sources. Specifically, we looked at four data sets: (1) the BABAM data set (*Robie et al., 2017*), in which measures were acquired from groups of flies behaving in open arenas, and variation came from the thermogenetic activation of 2381 different sets of neurons (the first-generation FlyLight Gal4 lines *Jenett et al., 2012*); (2) a *Drosophila* Genome Reference Panel (DGRP; *Mackay et al., 2012*) behavioral data set, in which measures were acquired in behavioral assays similar to the Decathlon experiments (sometimes manually, sometimes automatically), and variation came from the natural genetic variation between lines in the DGRP collection; (3) a DGRP physiological data set, in which measures are physiological or metabolic (e.g., body weight and glucose levels) and variation came from the natural genetic variation between lines in the DGRP collection; and (4) the split-Gal4 Descending Neuron (DN) data set (*Cande et al., 2018*) in which measures came from the same unsupervised cluster approach as *Figure 2*, and variation came from the optogenetic activation of specific sets of descending neurons projecting from the brain to the ventral nerve cord (*Namiki et al., 2018*). We analyzed these data sets with the same tools we used to characterize the structure of behavioral variation in the Decathlon experiments.

All of these data sets show substantial structure in their correlation matrices (*Figure 5A* and *Figure 5—figure supplement 1*). The BABAM and especially the DN correlation matrices contain numerous high correlation values, indicative of strong couplings between behaviors under these neuronal manipulations. The DGRP correlation matrices, especially the DGRP behavioral matrix, look more qualitatively similar to the Decathlon matrix, with lower, sparser correlations. This suggests that behavioral variation has coarsely similar structure whether variation arises intragenotypically (e.g., through stochastic variation in transcription; *Figures 3B and 1J*), intergenotypically among outbred individuals (*Figure 4E*), or intergenotypically among inbred lines derived from wild populations (*Figure 5A2*). A caveat of this conclusion is that sparse correlation matrices can arise either from true, biological independence of behavioral measures or from measurement error.

The connected components spectra of these matrices (*Figure 5B*) are similar in offering evidence of organization over a wide range of dimensionalities, including high dimensionalities. Only the BABAM spectrum has no power at the dimensionality of its raw count of measurements. The BABAM and DN spectra have a single predominant peak (at dimensionality = 1), suggesting that most measures belong to a single network of at least weak couplings. This is especially true in the BABAM data and is more true in the optogenetic experimental animals than controls in the DN data. The DGRP physiology data exhibits weak peaks at dimensionality = 1 and *d*, but also peaks at ~25 and 37 dimensions, suggesting that intergenotypic variation in physiology may have an intrinsic dimensionality in that range. Alternatively, since these data sets comprise multiple studies, organization at this dimensionality may reflect batch effects or multiple related measurements within studies. The spectrum of the DGRP behavior data looks similar to that of the Decathlons, with peaks at dimensionality = 1 and *d*, and evidence for structure over the full range of intermediate dimensionalities. There is also a peak at ~27 dimensions, which may also correspond to study-level effects. Ultimately, we found no evidence for low-dimensional organization in either DGRP data set. The distribution of individual lines in the space of DGRP behavior (and physiology) measures appears to be distributed single mode (*Figure 5C2 and C3*), like individual flies in the Decathlon (*Figure 1K*). In contrast, there is some organization of individual lines in the BABAM and DN data sets, likely reflecting neuronal perturbations affecting multiple circuit elements mediating the same behavior(s) (e.g., multiple lines targeting the same neuropil). Measures fall into clusters in all of these data sets except the DGRP behavioral measures (*Figure 5D*), which appear distributed around a single mode, perhaps reflecting the high dimensionality of behavior itself.

## Discussion

Individuals exhibit different behaviors, even when they have the same genotype and have been reared in the same environment. These differences might covary or lie on a manifold of specific geometry in

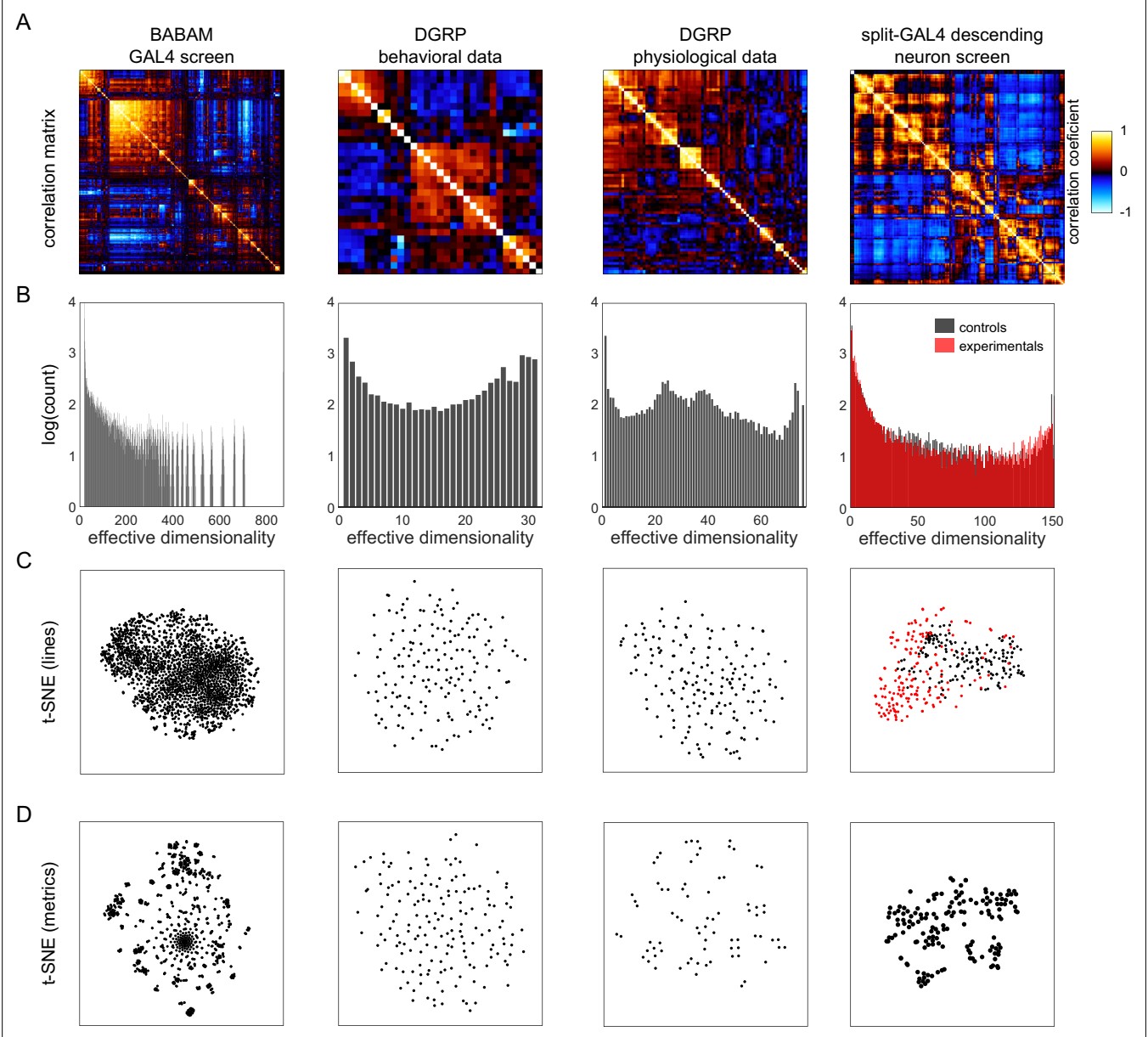

**Figure 5.** Analysis of *Drosophila* behavioral covariation in other non-isogenic populations. (**A**) Correlation matrices of previously published data sets. Rows correspond to analyses performed on each data set. From left to right, the data sets (columns) are as follows: line averages of supervised behavioral classifications following thermogenetic inactivation in the fly olympiad screen (*Robie et al., 2017*), line averages of behavioral phenotypic data from wild-type inbred lines in the *Drosophila* Genomic Reference Panel (DGRP) database, line averages of physiological phenotypic data from the DGRP database, line averages of the fold change in unsupervised behavioral classifications following optogenetic activation of descending neurons (*Cande et al., 2018*). (**B**) Connected components spectra for each correlation matrix (see Materials and methods). Color in the rightmost plots (**B–D**) indicates either control (Gal4 driver only) or experimental animals (Gal4 × dTrapA1). (**C**) Points corresponding to lines nonlinearly embedded using *t*-SNE from the D-dimensional raw measure space to two dimensions (from left to right, d = 871, 31, 77, 151). (**D**) Points corresponding to lines nonlinearly embedded using *t*-SNE from the *n*-dimensional raw measure space to two dimensions (from left to right, n = 2083, 169, 169, 176).

The online version of this article includes the following figure supplement(s) for figure 5:

**Figure supplement 1.** Structure of behavioral variation in non-Decathlon data sets.

behavioral variation space, but the structure of intragenotypic behavioral variation is uncharacterized. We designed a pipeline of 10 behavioral assays (*Figure 1*), which collectively yielded up to 121 behavioral measures per individual animal. We also used unsupervised clustering to identify an additional 70 measures per individual based on a time-frequency analysis of high-resolution video of the flies

behaving spontaneously (*Figure 2*). These measures were the fly-specific rates of exhibiting each of the 70 unsupervised behavioral modes. All in all, across three 15- day Decathlon experiments, we collected 191 behavioral measures from 576 flies. This allowed us to produce a full correlation matrix of the behavioral measures exhibited by inbred animals grown in the lab (*Figure 1E*).

There is a well-developed theoretical framework for understanding the multivariate correlation structure of phenotypes. In quantitative genetics, G-matrices characterize the variance and covariance structure of phenotypes (be they behavioral, physiological, morphological) stemming from genetic differences among individuals or strains (*Mackay, 2009*; *Bruijning et al., 2020*). These representations allow the quantitative prediction of responses to selection and constrain the combinations of phenotypes individuals can exhibit. As such, these representations are a key part of predicting the future trajectories of evolution. Just as the phenotypic variance can be parsed into genetic variance, environmental variance, GxE interaction variance, etc., covariance can be similarly dissected (*Charmantier et al., 2014*; *Berdal and Dochtermann, 2019*). For example, the classic model of phenotypic variance $V_P = V_G + V_E$ has a direct phenotypic covariance analogue: $Cov_P = Cov_G + Cov_E$. The last term ($Cov_E$) is further broken down into temporary environmental covariances and permanent environmental covariances ($Cov_{PE}$) that endure for the duration of observations (similar to our measurements of individual behavioral bias). In flies, we have the potential to directly measure $Cov_{PE}$ by rearing inbred animals in standardized lab environments, profiling their individual biases over a wide range of behavioral measures, and directly quantifying the variance and covariance of behavioral bias. This is significant for quantitative geneticists because a meta-analysis of behavioral traits indicates that across behaviors ~23% of variance can be attributed to heritable factors (*Dochtermann et al., 2019*). This means that environmental factors, which include both deterministic effects and stochastic intragenotypic effects, explain ~77% of behavioral variance. Thus, characterizing the structure of $Cov_{PE}$ will contribute to closing a significant gap in our understanding of the basis of behavioral diversity.

For ethologists and behavioral neuroscientists, this work yields a view of the geometry of intragenotypic behavioral variation, which can be thought of as an emergent product of developmental biological processes and the dynamic interaction of neural activity and animals' environment. From the full behavioral matrix, we made a so-called 'distilled' matrix in which any significant correlation indicates a surprising new relation between behaviors (*Figure 1G* and *Figure 1—figure supplement 11*). This form of the data minimizes duplicated measures of the same behavior, allowing us to cleanly analyze the geometry of behavioral variation. We found that behavioral measures were largely uncorrelated with each other, but small sets of behaviors were significantly correlated to varying degrees. This sparse correlation structure means that behavioral variation cannot be readily compressed to a small number of dimensions. Moreover, a single number cannot fully characterize the dimensional organization of a correlation matrix; so we developed a spectral approach (based on a connected components analysis of thresholded correlation matrices) that examined the degree of organization across all possible dimensionalities in the data (*Figures 1J, 2F, 4E and 5B*, *Figure 1—figure supplement 12*). This approach revealed organization across intermediate dimensionalities corresponding to correlations of varying strength between sparse sets of behaviors (*Figure 1F and H* and *Figure 1—figure supplement 12*). Embedding data points corresponding to individual flies from the high-dimensional space of individual biases into two dimensions produced a broad distribution around a single mode (*Figure 1K*), implying that there are no discrete types of flies.

One of the specific, surprising correlations we discovered was between 'clumpiness' and 'switchiness' (*Figure 1F, H* and *Figure 1—figure supplement 11*, *Figure 3—figure supplement 1*). These are slightly abstract, higher-order behavioral measures, corresponding respectively to the burstiness of turn/action/decision timing and the degree of independence between consecutive binary choices. We had no a priori reason to expect these measures would be correlated since one pertains to the structure of actions in time and one pertains to the persistence of trial-to-trial biases. However, their linkage may reflect a shared role in controlling the higher-order statistics of exploration. Sequences of behavior with clumps of bouts (either in time or in space) might contribute to fat-tailed distributions of dispersal that are advantageous over Brownian motion for foragers in environments of sparse resources (*Bartumeus et al., 2002*). Variation in switchiness and clumpiness across individuals might therefore reflect variation in multibehavioral navigational strategies, perhaps as part of bet-hedging evolutionary strategy (*Hopper, 1999*; *Kain et al., 2015*). From a perspective of biological mechanism, the correlation between these two behaviors (or other pairs we discovered) could be established

during development. Individual wiring (*Mellert et al., 2016*; *Linneweber et al., 2020*) or physiological variations in neurons that mediate more than one behavior could impart coupled changes to all such behaviors.

If such an explanation accounts for the correlation of clumpiness and switchiness, there may be shared neural circuit elements in the circuits controlling decision timing and decision bias. We tested this idea in a thermogenetic screen of circuit elements in the central complex, a brain region where heading direction is represented (*Seelig and Jayaraman, 2015*) in a ring-attractor circuit (*Kakaria and de Bivort, 2017*). We found that increasing the temperature changed the sign of the correlation between switchiness and clumpiness (*Figure 3—figure supplement 1D* and E), suggesting that changes to brain physiology can alter correlation structure, though these effects might instead be caused by any of the many changes in neural state that accompany a temperature change. Indeed, we also found that the effector-inducing temperature manipulation alone concomitantly changed clumpiness and switchiness in some lines. This suggests that potentially subtle alterations of circuit physiology (e.g., temperature shifts [*Haddad and Marder, 2018*] well within physiological limits) can affect the function of circuit elements governing multiple behaviors.

We included behavioral assays in the Decathlon pipeline under a number of constraints. The assays had to be high throughput, both in the number of flies that could be assayed and in measures being automatically acquired. Flies had to survive at high rates, and the measures had to be stable over multiple days (*Figure 1—figure supplement 1*), because the whole experiment lasted 15 days. Because not all behavioral measures showed robust stability for this duration (all showed at least some day-to-day stability), the distilled Decathlon matrices likely represent an underestimate of the behavioral couplings exhibited over short periods (and perhaps an overestimate of lifelong couplings as flies can live more than a month). In the end, we employed a number of spontaneous locomotion assays and simple stimulus-evoked assays like odor avoidance and phototaxis.

Light responses were measured in a number of assays (*Supplementary file 1*), specifically the LED Y-maze (*Werkhoven et al., 2019*; in which flies turned toward or away from lit LEDs in a rapid trial-by-trial format), the spatial shade-light assay (in which flies chose to stand in lit or shaded regions of an arena that only changed every 4 min), and temporal shade-light (in which the same luminance levels were used as the previous assay, but a fly experienced them by traveling into virtual zones that triggered the illumination of the whole arena at a particular luminance) (*Figure 1B*). These assays were potentially redundant, and we included this cluster of phototaxis assays in part as a positive control. However, the three phototactic measures we thought would be correlated a priori were, in fact, uncorrelated, each being represented in the distilled correlation matrix (*Figure 1G*). This may reflect flies using different behavioral algorithms (*Krakauer et al., 2017*), implemented by non-overlapping circuits, to implement these behaviors. Indeed, a lack of correlation between behavioral measures was the typical observation. This also suggests that we have not come close to sampling the full dimensionality of intragenotypic behavioral variation; if we were able to add more measurements to the experiment, they too would likely be uncorrelated.

To address potential biases in our sampling of assay and measure space, we performed an unsupervised analysis (*Berman et al., 2014*; *Cande et al., 2018*) of flies walking spontaneously in arenas. This approach has the potential to identify all the modes of behavior exhibited in that context. Moreover, because the unsupervised algorithm is fundamentally a clustering algorithm, it does not necessarily return a definitive number of clusters/behavioral modes (with more data, it can find increasingly more clusters). Because we can also extract second-order behavioral measures from this approach, such as Markov transition rates between modes, this approach has the potential to yield a huge number of measures. In the end, we were conservative in the number of measurements we chose to work with, matching it to the same order of magnitude as the number of flies we tested. The correlation matrix for the unsupervised behavioral measures featured stronger correlations than the distilled Decathlon matrices (*Figure 2E*). Yet it had a similar connected components spectrum, indicating many dimensions of variation and that not all behavior modes had been sampled (*Figure 2F*).

Interestingly, the blocks of structure in the correlation matrix aligned, to some extent, with the blocks of structure in the Markov transition matrix of these behavioral measures. This suggests that behaviors mediated by non-overlapping circuits (those that vary independently across individuals) more rarely transition to each other over time. Conversely, behaviors mediated by overlapping circuitry are likely to follow each other sequentially. This may reflect the influence of internal states

(*Calhoun et al., 2019*), with an internal state jointly determining what subset of overlapping neurons drives behaviors that are appropriate to string together in succession (e.g., *Seeds et al., 2014*). We did not assess the day-to-day persistence of behavioral modes identified in the unsupervised analysis, so the observed variation across flies could reflect moods rather than permanent personalities. Indeed, *Hernández et al., 2020* find that high-level clusters in unsupervised modes identified through an information bottleneck analysis of the behavioral transitions that occur over many minutes align with clusters in the cross-individual covariance matrix. This suggests either that individuality observed in approximately hours -long video recordings reflects approximately hours -long transient states or, alternatively, that the structure of days-long or lifelong variation mirrors that of hours-long variation. The latter may be particularly plausible, given that previous studies that measured spontaneous microbehaviors using supervised (*Kain et al., 2013*) and unsupervised (*Todd et al., 2017*) approaches over repeated tests found substantial day-to-day correlations.

We investigated whether individual variation in transcript abundance would predict individual behavioral variation (*Figure 3*). At the end of the Decathlon, flies were flash-frozen and their heads were RNA sequenced. We fit linear models for thousands of gene-behavior measure pairs to identify a set of genes that predict behavioral variation. Some measures had many gene predictors, while others had none. Gene function enrichment analysis revealed that variation in the expression of genes involved in respiration and other kinds of metabolism predicted variation in many behavioral measures. Genes involved in neuronal and developmental processes were also enriched among predictors of individual behavior. These two functional categories are likely linked as there are strong causal couplings between metabolism and neural activity (*Mann et al., 2020*). Variation in behavioral measures without expression correlates may not have mechanistic origins in transcriptional variation, or the genes that do determine these behaviors may be not expressed in adults, expressed at low levels, or expressed in too few cells to detect in bulk head tissue.

Increasing transcriptional variation by adding genetic variation had the potential to change the correlation structure of behavioral measures. To our surprise, the distilled correlation matrix of outbred flies was qualitatively similar to that of our original inbred Decathlon (*Figure 4*). Both outbred and inbred matrices were dominated by independent axes of variation and sparse correlations between axes, with rough agreement in the specific pairwise correlations between these two data sets. These two data sets may have differed in their absolute variances (while appearing qualitatively similar in their covariances), but the normalization steps we took to resolve inter-Decathlon and assay batch effects precluded easy assessment of this possibility. That outbred and inbred variation had qualitatively similar structures raises the possibility that the same kinds of biological fluctuations underlie behavioral variation in populations of each of these kinds. The high dimensionality of the inbred and outbred matrices suggests that behaviors can evolve largely independently of one another (i.e., without pleiotropic constraint) either via plastic mechanisms within a genotype or by natural selection in a genetically diverse population.

We finally examined the structure of behavioral variation in collections of flies where variation came from three additional sources: thermogenetic activation of 2205 sets of neurons across the brain (*Robie et al., 2017*), optogenetic activation of 176 sparse populations of descending neurons connecting the brain to the motor centers of the ventral nerve cord (*Cande et al., 2018*), and variation in genotype across ~200 inbred strains derived from wild flies (*Mackay et al., 2012*). Overall, all of these data sets exhibited high degrees of independence in their behavioral measures, though there is some variation from one data set to the next. This variation could arise from several non-biological sources, including differing extents of collecting multiple measures from the same experiment, which can inflate correlations due to coupled noise; differences in the number of repeated or numerically coupled measures; batch effects from combining studies conducted in different labs at different times; and varying signal-to-noise ratios in the measurements. The structure of behavioral variation in the neural activation data sets was somewhat different from that of inbred and outbred flies, with a dimensionality smaller than that of the number of measures, and substantially more organization in lower dimensionalities (*Figure 5A and B*). These data sets also showed clustering of individual flies in behavior space (*Figure 5C*). Behavioral variation across the inbred strains derived from wild flies was organized qualitatively similarly to the variation across individual flies in inbred and outbred populations, again suggesting that the biological fluctuations across genotypes mirror those within a genotype as respects the coordination of behavior.

This work represents the most complete characterization to date of the structure of behavioral variation within a genotype. We found that there are no discrete types of flies, and there are many independent dimensions of behavioral variation. Moreover, the similar organization of biological variation within and among genotypes suggests that fluctuations in the same biological processes underpin behavioral variation at both of these levels. Elucidating the causal molecular and genetic fluctuations underpinning intragenotypic variation and covariation will be a challenging and illuminating future research direction.

## Materials and methods

### Software

All software for analysis and figures in this paper as well as documentation are available at http://lab.debivort.org/structure-of-behavioral-variability/. Static DOI for this code is available at https://doi.org/10.5281/zenodo.4110049. Actively maintained versions of the analysis software are available at https://github.com/de-Bivort-Lab/decathlon (copy archived at swh:1:rev:6c9e338db6f03e42bbbb2e-2afa0cfd52162e7772; *de Bivort, 2021a*).

MARGO (animal tracking and stimulus delivery software): https://github.com/de-Bivort-Lab/margo (copy archived at swh:1:rev:fe61a873494e464d2a3ee48f67e885eb95359e0a, *de Bivort, 2021b*).

Decathlon Data Browser (interactive web application for exploring data): .http://decathlon.debivort.org.

### Data

Data such as behavioral measures, unsupervised embedding PDFs, RNAseq read counts, and formatted versions of the BABAM, DGRP phenotypic, and DN screen data are available at http://lab.debivort.org/structure-of-behavioral-variability/. Static DOI for these data in addition to the unsupervised classification embedding time series data is available at https://doi.org/10.5281/zenodo.4110049. Our raw tracking data from the assays covers ~600 flies over 12 days of continuous tracking at between 3 and 60 Hz and are greater than 50 GB in size. Our unsupervised classification videos comprise ~144 million frames of uncompressed video data and are greater than 8 TB in size. For these reasons, we have not posted the rawest forms of the data to online repositories and instead offer them upon request by external hard drive.

### Fly strains

All Decathlon experiments were performed on virgin female fruit flies derived from inbred wild-type Berlin-K (BSC# 8522) isogenic flies or custom outbred flies (NEX) (REFEF). Prior to Decathlon testing, a lineage of Berlin-K isogenic flies (Berlin-K$^{iso}$) were selected for robustness from a pool of inbred lineages as the result of a short screen designed to mimic the stresses of the Decathlon assay battery. All inbred lineages tested were derived from three wild-type strains: Berlin-K (BK), Canton-S (CS), and Oregon-R (OR). For each wild-type strain, six virgin females were dispensed into separate vials and paired with a single male fly to establish separate lineages. Parental flies were removed from their vials after 2 days to separate them from their progeny. For all successive generations after the first generation, the six virgin females were picked from the three vials (two from each) with the highest number of progeny to select for fecundity and overall ease of maintenance. Virgins were then dispensed into separate vials with full sibling males for inbreeding. After 13 generations of inbreeding, this way the resulting lineages (six total for each strain) were screened for overall activity level via behavioral testing on the Y-maze assay in cohorts of 120 flies from each lineage. For each strain, the lineage with the highest number of choices (turns) in the Y-maze from was kept for further testing to improve overall sampling in behavioral experiments, resulting in a single lineage from each strain: Berlin-K$^{iso}$, Canton-S$^{iso}$, and Oregon-R$^{iso}$.

To screen robustness to the stresses of multiple days of behavioral testing such as periodic food deprivation and repeated cold anesthetization, cohorts of 96 flies from the resulting inbred lines were singly housed and repeatedly tested on the Y-maze assay over a 3 -day period. Berlin-K$^{iso}$ and Canton-S$^{iso}$ were selected as the lines with lowest and second lowest (respectively) mortality after 3 days of testing. To potentially introduce hybrid vigor and reduce the deleterious effects of inbreeding, four F$_1$ hybrid inbred strains were generated by crossing each combination of parental sex and inbred strain

(e.g., ♂ Berlin-K$^{iso}$ × ♀ Canton-S$^{iso}$, ♀ Berlin-K$^{iso}$ × ♂ Canton-S$^{iso}$, etc.) and screened for mortality in the 3-day experiment described above. Contrary to our expectation that increased heterozygosity in the $F_1$ hybrid lines would result in more robust flies and lower mortality, $F_1$ flies displayed intermediate mortality and activity level (choice number) between Berlin-K$^{iso}$ and Canton-S$^{iso}$ parentals. Thus Berlin-K$^{iso}$ flies were ultimately selected for the Decathlon screen.

## Fly handling

Parental flies and Decathlon flies (prior to individual housing) were raised on CalTech formula cornmeal media under 12 hr/12 hr light and dark cycle in an incubator at 25 °C and 70% humidity. A single cohort of virgin Decathlon flies were collected over an 8 hr period (10 AM to 6 PM) and stored in group housing up to a maximum of 288 flies. On the following day, flies were anesthetized using carbon dioxide ($CO_2$) and were aspirated into custom multiwell plates for individual housing (FlySorter LLC, Seattle, WA) on cornmeal media. Individually housed flies were stored in a custom imaging box for circadian activity measurements (circadian chamber) on a 12 hr light and dark cycle (10 AM to 10 PM), which was housed inside of an environmentally controlled room at 23 °C and 40% humidity. Food media was replaced and individual housing trays were cleaned every 2 days while flies were in behavioral testing to keep food moisturized and prevent microbial buildup.

Following the start of Decathlon (3 days post-eclosion), flies were removed from the circadian chamber each day for behavioral testing between 11 AM and 2 PM. Prior to all behavioral experiments (excluding olfaction), flies were cold anesthetized by transferring individual housing plates and food trays to an ice-chilled aluminum block in a 4 °C for 10 min. This time was the minimum time required to reliably induce chill coma in most flies given the mass and low thermal conductivity of the cornmeal media separating the flies and the chill block. Once anesthetized, flies were individually transferred via aspiration to room temperature custom behavioral arenas (without food) where they quickly awoke (typically 1–10 s) after transfer. After all flies were transferred, they were given an additional 20 min to recover prior to behavioral testing. Flies were then tested in custom behavioral platforms, lasting as little as 15 min (olfaction only) and as much as 2 hr (typical). Following testing, flies were returned to individual housing via aspiration either directly (olfaction only) or after re-anesthetizing by transferring the custom behavioral arenas to an ice-chilled aluminum block for 10 min. Once all flies were returned, the individual housing plates were returned to the circadian chamber between 2 and 5 PM.

## Custom behavioral experiments

### Assay fabrication

Unless otherwise specified, all tracking was conducted in custom imaging boxes constructed with laser-cut acrylic and aluminum rails. Schematics of custom behavioral arenas and behavioral boxes were designed in AutoCAD 2014 (AutoDesk). Arena parts were laser-cut from black and clear acrylic and were formed as stacked layers joined with Plastruct plastic weld. Arena floors were made from sandblasted clear acrylic and arena lids were made from custom-cut clear eighth inch acrylic. Schematics for behavioral boxes and behavioral arenas can be found on the de Bivort Lab schematics repository (https://github.com/de-Bivort-Lab/dblab-schematics). Illumination was provided by dual-channel white and infrared LED array panels mounted at the base (Part BK3301, Knema LLC, Shreveport, LA). Adjacent pairs of white and infrared LEDs were arrayed in a 14 × 14 grid spaced 2.2 cm apart. White and infrared LEDs were wired for independent control by MOSFET transistors and a Teensy 3.2 microcontroller. Two sand-blasted clear acrylic diffusers were placed in between the illuminator and the behavioral arena for smooth backlighting.

For circadian experiments, flies were housed and imaged in individual fly storage units (FlyPlates) from FlySorter, LLC. Circadian imaging boxes consisted of a small (6" × 6" × 18") enclosed behavioral box with a dual-channel LED illuminator on a 12:12 hr light and dark cycle programmatically controlled via MARGO behavioral tracking software (https://github.com/de-Bivort-Lab/margo). A heat-sink was affixed to the underside of the illuminator panel (outside of the box), and fans were used to ensure the interior of the boxes remained consistent with the temperature of the environmental room.

The olfactory sensitivity assay was performed in a custom behavioral chamber modeled on *Claridge-Chang et al., 2009*. The apparatus consisted of 15 parallel tunnels constructed from Accura 60 plastic using stereolithography (In'Tech Industries) fabrication. Stainless steel hypo tubing (Small Parts) was used to connect the apparatus with (ID: 0.7 mm). Odorized or clean air

was delivered via Teflon odor tubes to inlet ports at each end of the tunnel and streams vented to the room through exhaust ports in the center forming a sharp choice zone. An active vacuum was not applied to the exhaust ports, and the tunnels operated close to atmospheric pressure throughout the experiment. A clear acrylic lid was clamped in place above the apparatus to ensure an air-tight seal during odor presentation. Air dilutions could be made independently for each side of the apparatus. A custom 15-way PEEK manifold was used on each side to split the odorized flow equally between 15 tunnel inlets. A final valve (SH360T041; NResearch) was used immediately upstream of each manifold to quickly switch between pure dehumidified air and the odorized stream.

Phototactic stimuli were delivered with a custom 12" × 12" PCB designed in Express PCB CAD software and manufactured by ExpressPCB. Briefly, the PCB was designed to power and independently control 216 white light LEDs (11.34 mW ± 4.3 µW at 20 ma) geometrically arranged to match the maze arm ends of an array of Y-shaped behavioral arenas. LED intensity control was provided over USB via board interface with a Teensy 3.2 microcontroller and constant current driver boards (Adafruit 24-Channel TLC5947 LED Driver). Holes in the footprint shape of each individual arena were custom cut into the PCB with water jet cutter to provide infrared backlighting.

Visual stimuli in the spatial shade-light, temporal shade-light, and optomotor assays were delivered in a behavior box modified to accommodate stimulus delivery with an overhead projector (Optoma S310e DLP). The DLP color wheel was removed to reduce apparent flicker to the flies caused by low sensitivity to red light in flies. Stimuli were accurately targeted to individual flies by creating a registration mapping between the projector and the tracking camera using previously described method using MARGO tracking software. Briefly, a 2D polynomial registration model of a projector targeting coordinates of a small (10 px) was fitted to coordinates obtained by tracking the dot as it was rastered over the projector display field. All visual stimuli were crafted and displayed with PsychToolbox.

Single-fly videos used in the motionmapper unsupervised classification pipeline were acquired in behavioral boxes as described above. We replaced the dual-channel LED illuminator with an LCD screen backlight to reduce apparent non-uniformities in the videos. Behavioral arenas consisted of custom thermoformed clear PTEG lid with sloping sides to prevent wall walking. Lids were coated with sigmacote (Sigma-Aldrich SL2-100ML) to prevent flies from walking on the ceiling and were clamped in place to a 0.25" tempered glass base via custom acrylic 2 × 2 array of arenas.

## Behavioral tracking

Flies were imaged with overhead tracking cameras at varying resolutions and frame rates. Unless otherwise specified, images for behavior tracking were acquired at 10 Hz with a 1280 × 1024 pixel BlackFly GigE camera (PointGrey BFLY-U3-13S2M-CS) fitted with a Fujinon YV2.8 × 2.8 SA-2, 2.8– 8 mm, 1/3", CS mount lens. Circadian and optomotor experiments were conducted with 3 Hz and 60 Hz imaging, respectively. Tracking images for the odor sensitivity assay were acquired at 1328 × 1048 pixel at 23 Hz (PointGrey FMVU-13S2C). Single-fly videos for unsupervised classification were acquired.

Fly tracking and stimulus control for all experiments (excluding unsupervised single-fly imaging) were programmed in MATLAB and implemented with MARGO (*Werkhoven et al., 2019*). The MARGO tracking algorithm has been previously described in detail. Briefly, binary foreground blobs were segmented from a thresholded difference image computed by subtracting each frame from an estimate of the background. A rolling model of the background was computed as the median image of the rolling stack of background sample images. Every 2 min, a new sample image of the background was acquired and replaced the oldest image in the stack. Regions of interest (ROIs) were defined to encompass only a single behavioral arena (i.e., one fly), and the tracking algorithm proceeded independently for each ROI. The tracking threshold used to segment binary foreground blobs from the background was manually set prior to tracking and was independent for each experiment. Blobs below a minimum or maximum area (also defined manually for each experiment) were excluded from tracking in each frame. After filtering, the centroid trace of each ROI was updated with the position of the blob with shortest distance to the last known (non-NaN) position of each trace. The centroid trace of any ROI with no detected blobs in any given frame received no position (i.e., NaN). Output measurements of speed and area were converted from pixels/s and pixels$^2$ to mm/s and mm$^2$, respectively, by measuring the length of a known landmark size (e.g., arena diameter) prior

to tracking. Fisheye distortion of camera lenses was modeled and corrected via MATLAB's camera calibrator app.

For the acquisition of single-fly videos used in the motionmapper unsupervised classification pipeline, flies were tracked in real time to reduce video file size by extracting a 200 × 200 pixel region around the fly centroid. Tracking was performed by computing the centroid of the entire difference image acquired and was calculated via the method described above. Tracking was coordinated simultaneously for four 1280 × 1024 pixel resolution cameras (PointGrey CM3-U3-13Y3M-CS) at 100 Hz with a custom LabView script obtained from the J. Shaevitz lab (personal communication), which was lightly modified to add background subtraction to the tracking.

## Decathlon experimental design

Flies underwent behavioral testing from 3 days post-eclosion up to a maximum of 14 days post-eclosion depending on when unsupervised classification was performed. For each Decathlon experiment, behavioral testing began with activity screen of 288 flies in the Y-maze assay. Flies were then sorted by number turns made during the activity screen. The top 192 flies with the highest activity level were selected for testing in the remainder of the Decathlon and were transferred to individual housing on day 3 post-eclosion for overnight circadian behavior profiling. On subsequent days, all flies were anesthetized as described above and transferred from individual housing into a behavioral assay each day (from days 3 to 11 post-eclosion) and were returned individual housing for overnight circadian behavioral measurement. Flies were tested in cohorts dependent on the throughput capacity (described for each assay below) of the assay run on any given day. Although the range of testing times for all animals on some days was relatively broad (all experiments were conducted between 11 AM and 5PM), no behavioral experiment lasted more than 2 hr in duration, meaning that individual cohorts typically spent more than 3 hr (2 hr experiment +2× anesthetization and transferring) off of food media. The order of individual assays (shown in *Figure 1*) was randomized at the start of each Decathlon with the exception of activity screening and unsupervised imaging, which occurred at the beginning and end of each Decathlon (respectively) for logistical feasibility. Descriptions and implementation details of the Decathlon behavioral assays are provided below and are summarized in *Supplementary file 1*.

After day 11 post-eclosion, remaining living flies were split into three roughly equal-sized cohorts (approximately 56 flies each) for unsupervised behavioral imaging, which were imaged in groups of four flies between 10 AM and 6 PM on subsequent days. Cohorts of flies were flash-frozen on liquid nitrogen immediately following unsupervised behavioral imaging and prepped for RNAseq as described above.

## Arena circling and circadian assays

Both the arena circling and circadian assays consisted of exploration of a circular arena. Although a near identical list of measures of locomotor handedness, speed, and movement bout dynamics were recorded in each assay (circadian includes gravitaxis also), the assays are distinguished by their background light level, duration, behavioral arena, and access to food. The arena circling assay was conducted over 2 hr with constant light in a wide (28 mm diameter) and shallow (1.6 mm depth) arena without food. The circadian assay was conducted overnight (20–21 hr) on a 12:12 hr light and dark cycle (10 AM:10 PM) with each fly in a narrow (6.8 mm) and deep (10.6 mm) well of a 96-multiwell plate. The depth of the arena in addition to the difference in arena ceiling (sigmacoted acrylic in arena circling and uncoated plastic mesh circadian) adds a vertical and body orientation dimension to the circadian assay not present in arena circling. This feature not only adds a measure of floor vs. ceiling preference (i.e., gravitaxis) to the circadian assay but also potentially confounds measures of circling directionality due to the fact that flies can circle the arena right-side up on the floor, upside-down on the ceiling, or sideways on the walls.

## LED Y-maze assay

Individual flies explored symmetrical Y-shaped arenas with LEDs at the end of each arm. For all arenas in parallel, real-time tracking detected which arm the fly was in at each frame. At the start of each trial, an LED was randomly turned on in one of the unoccupied arms. Once flies walked into one of these two new arms, all the LEDs in the arena were turned off and a new trial was initiated by randomly

turning on an LED in one of the newly unoccupied arms. This process is repeated for each fly independently over 2 hr. Turns were scored as positively (toward a lit LED) or negatively (toward an unlit LED) phototactic. LEDs were either fully off (OFF) or were driven at maximum intensity (ON). No intermediate LED intensity values were tested. In addition to the phototactic bias, measures of locomotor handedness peed, choice timing, and choice sequence were also recorded.

## Odor sensitivity assay

In the odor sensitivity assay, flies explored linear tunnels where half of the tunnel contained an odor (typically noxious) and the other contained no odor. Odor ports positioned at either end of the tunnels delivered either odor or odorless air. Vents positioned at the midpoint of each tunnel form a sharp boundary between the two air streams. Flies were presented with a 3 min baseline period with no odor on either side followed by a 12 min experimental period with half odor and half no odor. Sensitivity to the odor was estimated by measuring the fraction of occupancy time spent in the odor half of the tunnel. Flies showed a weak aversion to the designated odor side during baseline measurement, prior to the presentation of odor ($0.47 \pm 0.30$ mean pre-odor occupancy). Flies displayed strong initial aversion ($0.17 \pm 0.22$ mean first half odor occupancy) to the side of the tunnels with the odor, which slightly diminished ($0.22 \pm 0.23$ mean second half occupancy) over the stimulus duration. In addition to odor occupancy, locomotor handedness was scored as the average direction of heading direction reversals on the long axis of the tunnels. In addition to the above measures, speed, turn timing, and turn direction sequence were also recorded.

## Optomotor assay

In this assay, optomotor stimuli were centered on the bodies of flies by projecting them onto the floor of their arenas in which they are walking freely. These stimuli consisted of a maximally contrasting (black = 0, white = 1, min. power = 79 µW ± 61 nW, max. power = 2.4 mW ± 4.7 µW) rotating pinwheel (spatial frequency = 0.18 cycles/°, rotational speed = 320 °/s) and typically evoked a turn in the direction of the rotation to stabilize the visual motion. The pinwheel center followed the position of the fly as it moved so that the only apparent motion of the stimulus is rotational motion around the body. Therefore, the stimulus was closed loop with respect to position and open loop with respect to rotation velocity. We observed that optomotor responses could be reliably elicited, provided individuals were already moving when the pinwheel was initiated. We therefore only presented the pinwheel when (1) flies were moving, (2) a minimum inter-trial interval (2 s) had passed to prevent behavioral responses from adapting, and (3) flies exceeded a minimum distance from the edge of the arena to ensure that the stimulus occupied a significant portion of the animal's FoV. Over 2 hr of testing, an optomotor index was calculated for each fly as the fraction (normalized to [–1,1]) of body angle change that occurred in the same direction as the stimulus rotation over the duration of the stimulus. On average, flies displayed reliable optomotor responses (mean = $0.49 \pm 0.16$) when stimulated. In addition to optomotor index measures of fly activity, movement bout dynamics and locomotor handedness were also recorded.

## Spatial light-shade assay

Ambient light preference is estimated in this assay by projecting a circular stimulus with one side fully bright (intensity = 1) and the other side fully dark (intensity = 0), with the two regions separated by a small (10% of the arena diameter) boundary zone of intermediate brightness (intensity = 0.5) (min. power = 79 µW ± 61 nW, max. power = 2.4 mW ± 4.7 µW). A light occupancy measure was calculated for each fly as the fraction of time spent in the lit region divided by the total time spent in both the lit and unlit regions. Time spent in the intermediate boundary zone was scored as no preference and was excluded from the calculation. Each 2 hr experiment was divided into 15 stimulus cycles with each cycle consisting of an alternating 4 min baseline block where each arena was unlit and a 4 min experimental block where shade-light stimuli were targeted to each arena. At the start of each block, each stimulus was rotated to center the intermediate zones (or virtual intermediate zone in the baseline block) over the flies' bodies. In effect, this required flies to move from the intermediate zone during the block to express any preference. In addition to light occupancy measures of fly activity, movement bout dynamics and locomotor handedness were also recorded.

## Temporal light-shade assay

Ambient light preference is estimated in this assay by projecting a fully bright (intensity = 1) or fully dark (intensity = 0) stimulus to the entire arena when flies crossed an invisible virtual boundary in the center (min. power = 79 μW ± 61 nW, max. power = 2.4 mW ± 4.7 μW). Therefore, this stimulus paradigm was distinct from the spatial light-shade assay in that flies could not express a preference by navigating with respect to an apparent spatial pattern of light. A light occupancy measure was calculated for each fly as the fraction of time flies received the lit stimulus divided by the total duration of the experiment (2 hr). The invisible boundary zone was always positioned in the center of the arena such that a randomly exploring fly would receive both stimuli in roughly equal amounts. The angular orientation of the boundary zone was randomly initialized for each arena independently at the start of the experiment. To avoid rapid switching or flickering of the stimulus for flies sitting directly on the boundary, flies were required to cross a small buffer zone (5% of the arena diameter) beyond the arena center into the other zone before switching the stimulus.

## Y-maze assay

In the Y-maze assay, locomotor handedness was estimated as flies explored symmetrical Y-shaped arenas. Each time flies changed position from one arm of the maze to another, turns were scored as left-handed or right-handed depending on whether the chosen arm was to the left of right on the choice point (i.e., the arena center). To avoid scoring centroid estimation errors around the choice point or small forays into new arms as turns, the flies were required to traverse a minimum distance (60% the length of the arm) into any arm before a turn was scored. In addition to locomotor handedness, measures of speed, movement bout dynamics, turn choice timing, and turn choice sequence were also recorded.

## Central complex thermogenetic screen

We extracted measure of turn timing clumpiness (choiceClumpiness) and turn direction switchiness (handSwitchiness) from a previously conducted thermogenetic screen for light-dependent modulation of locomotor handedness (*Skutt-Kakaria et al., 2019*). Gal4 driver and neuronal effector parental lines were crossed to generate $F_1$ progeny with sparse neuronal expression of effectors that could be thermogenetically activated (dTrpA1) or silenced (Shibire$^{ts}$). For all screen experiments, fly locomotor handedness was assayed for 4 hr in the Y-maze with the following temperature program: 1 hr at 23 °C (permissive temperature), 1 hr temperature ramp up to the restrictive temperature (Shibire$^s$ = 29 °C, dTrpA1 = 32 °C), 1 hr at the restrictive temperature, and 1 hr temperature ramp down from the restrictive temperature to 23 °C. Although activity level varied over the duration of the experiment, flies made many turns throughout all temperature blocks: completing an average of 193 ± 119 and 757 ± 311 turns in the permissive and restrictive periods, respectively. The background light was repeatedly switched on and off in time blocks (min = 5 s, other times = 10 and 30, max = 60 s, each hour had the same repeated sequence) in a random order shared between all screen experiments. Because we were interested only in temperature-dependent modulation of turn dynamics, we computed individual clumpiness and switchiness scores for all turns within each temperature condition (permissive and restrictive) regardless of the light condition.

## RNAseq and transcriptomic analyses

Single-fly RNAseq preparation and sequencing was performed using TM3'seq, a high-throughput low-cost RNA protocol previously described in *Pallares et al., 2020*. An overview of the method is provided below, but more detailed descriptions are available in an online protocol and in the original publication.

### Fly tissue preparation

Following Decathlon behavioral testing, flies were briefly anesthetized on $CO_2$ and transferred to a 96-well plate via aspiration. Immediately following, flies were flash-frozen on liquid nitrogen and were decapitated to separate heads and bodies. Well plates were then transferred to a dry ice and ethanol cold bath to keep the samples frozen while heads were individually transferred to a separate 96-well plate with a cold probe needle. Samples were then stored at –80 °C. After storage, tissue was ground in 100 μl of lysis buffer with a 2.8 mm stainless steel grinding bead steel grinding bead for 10 min at

maximum speed with a homogenizer. CyBio FeliX liquid handling robot (Analitik Jena) was used to perform mRNA extraction from the resulting lysate using a Dynabeads mRNA DIRECT Purification Kit (ThermoFisher, #61012) and a custom protocol (*Pallares et al., 2020*) optimized for low cost, yielding approximately 10–20 ng mRNA per head.

## Library preparation

RNA (10 µl of 1 ng/µl mRNA) was added to 1 µl of 0.83 µM oligo (Tn5ME-B-30T) for a 3 min incubation at 65 °C immediately prior to reverse transcription. The first strand of cDNA was synthesized by reverse transcription of the mRNA via a 1 hr incubation at 42 °C by adding the following to the reaction mixture above: 1 µl SMARTScribe RT (Takara, #639538), 1 µl dNTPs 10 mM (NEB, #N0447S), 2 µl DTT 0.1 M (Takara, #639538), 4 µl 5× First-Strand buffer (Takara, #639538), and 1 µl B-tag-sw oligo. Following synthesis, the reverse transcriptase was inactivated via a 15 min incubation 70 °C. A cDNA amplification mixture was prepared by adding 5 µl of the resulting first-strand cDNA with 7.5 µl of OneTaq HS Quick-load 2× (NEB, #M0486L) and 2.5 µl water. The cDNA was then amplified in a thermocycler via the following program: 68 °C 3 min, 95 °C 30 s, [95 °C 10 s, 55 °C 30 s, 68 °C 3 min] * 3 cycles, 68 °C 5 min.

Tn5 tagmentation was used alongside universal adaptors for library amplification. The adapter-B was previously added during synthesis of the first cDNA strand. To create a Tn5 adaptor-A, an adapter annealing mixture was prepared by adding 10 µl (100 µM) of a forward oligo (adapter-A) and 10 µl (100 µM) reverse adapter-A oligo (Tn5MErev) to 80 µl re-association buffer (10 mM Tris pH 8.0, 50 mM NaCl, 1 mM EDTA). Oligos were annealed in a thermocycler with the following cycle program: 95 °C for 10 min, 90 °C for 1 min followed by 60 cycles reducing temperature by 1 °C/cycle, hold at 4 °C. 5 µl of 1 µM annealed adapter was then anneal to 5 µl of Tn5 in a thermal cycler for 30 min at 37 °C. 5 µl of cDNA was mixed with 1 µl of pre-charged Tn5. The adapter-A loaded Tn5 was diluted 7× in re-association buffer: glycerol (1:1), 4 µl of TAPS buffer 5× pH 8.5; 50 mM TAPS, 25 mM MgCl$_2$, 50% v/v DMF, and 5 µl of water. The mixture was then and incubated for 7 min at 55 °C followed by an additional 7 min incubation with 3.5 µl of SDS 0.2% (Promega, #V6551) to ensure that Tn5 was dissociated from the cDNA. The resulting cDNA libraries were then amplified. Briefly, 10 µl of OneTaq HS Quick-Load 2× (NEB, #M0486L), 1 µl i5 primer 1 µM, 1 µl i7 primer 1 µM, and 7 µl of water were used to amplify 1 µl of the tagmentation reaction following the program: 68 °C 3 min, 95 °C 30 s, [95 °C 10 s, 55 °C 30 s, 68 °C 30 s] * 12 cycles, 68 °C 5 min.

## Sequencing and expression quantification

Samples were sequenced on an Illumina HiSeq 2500 in separate runs using dual indexes and single-end ~100 bp sequencing. Low-quality bases and adapter sequences were removed from reads using Trimmomatic 0.32 (SE ILLUMINACLIP: 1:30:7 LEADING: 3 TRAILING: 3 SLIDINGWINDOW: 4:15 MINLEN:20; *Bolger et al., 2014*). Downstream analysis was only performed on reads at least 20 nucleotides after trimming. Reads were then mapped to the r6.14 *Drosophila melanogaster* genome assembly using STAR (*Dobin et al., 2013*), and were assigned to genes using featureCounts from the package Subread (*Liao et al., 2014*). Read counts were further filtered to include only reads assigned to protein coding genes with a minimum of 500k reads per individual (n.b. measuring 3′ biased expression).

## Enrichment analyses

Following sequencing, raw reads from individual flies with fewer than 1M reads total were removed. Raw read counts were normalized to reads per million (RPM) and were further filtered to exclude genes with less than 10 RPM averaged across all remaining flies. Individual fly read distributions were then quantile normalized so that the distributions of normalized reads were matched across all individual samples. One-dimensional linear models were fit using expression data from the remaining 6710 genes to predict each of the 97 individual behavioral scores (97 × 6710 models total). 69 × 6710 models were fit for unsupervised measures. Gene lists were constructed for each behavior separately by collecting their respective significantly predictive genes (model p-value < 0.05).

Enrichment scores were computed using clusterProfiler (*Yu et al., 2012*) as part of the Bioconductor (https://www.bioconductor.org) package for R (version 3.6; *R Development Core Team, 2020*). We performed KEGG enrichment analysis on predictive gene lists from each behavior independently

to identify functional categories of genes overrepresented for any particular behavioral measure compared to the background list of 6710 genes. clusterProfiler uses a Benjamini–Hochberg procedure (**Benjamini and Hochberg, 1995**) to adjust p-values. We report these adjusted p-values. KEGG categories were defined as significant if the adjusted p-values for the category were less than 0.05. Significant categories and their corresponding genes were identified separately for each behavior. Significant category (and gene) lists for groups of behaviors (i.e., all behaviors or a priori group behaviors) were constructed as the union of the significant enrichment categories (and genes) from each group's member behaviors. To assess reproducibility of enrichment categories and their corresponding gene sets, enrichment lists were repeatedly generated from bootstrap resampled individuals (500 replicates), downstream of quantile normalization. Average p-values were calculated as the mean across bootstrap replicates of the minimum adjusted p-value for each enrichment category across behaviors within a group.

## Genomic sequencing of inbred flies

DNA was extracted from individual flies using Zymo Quick-DNA Microprep Kit (Cat# D3020). Sequencing libraries were prepared using the transposase Tn5 using the same protocol described above (from cDNA). Samples were sequenced on an Illumina HiSeq 2500 using dual indexes, paired-end ~150 bp sequencing. After de-multiplexing, low-quality bases and adapter sequences were removed from reads using Trimmomatic 0.32 (SE ILLUMINACLIP: 1:30:7 LEADING: 3 TRAILING: 3 SLIDINGWINDOW: 4:15 MINLEN: 20). Reads were then mapped to the r6.14 *Drosophila melanogaster* genome assembly using BWA -aln (**Li and Durbin, 2009**). Genotypes were generated following GATK best practices (https://software.broadinstitute.org/gatk/best-practices/; **McKenna et al., 2010**). Heterozygosity was estimated using VCFTools v0.1.16 `--het` (**Danecek et al., 2011**).

## Quantification and statistical analysis

### Merging Decathlon data sets

Inspection of raw measure distributions separated by batch (i.e., a single cohort of flies tested in a single behavioral box at the same time) showed batch effects on the sample means and dispersion even within the same Decathlon experiment. Therefore, data was z-scored separately by batch and measure type to control for sample differences in mean and variance. Data of the same assay, measure, and day of testing (i.e., the unique measures from a single day of Decathlon behavior) were then combined, resulting in an initial $n \times d$ data matrix for a single Decathlon experiment, where $n$ is the number of individuals and $d$ is the number of measures. The matrix contained a substantial fraction of missingness (mean = 0.23 ± 0.22) within measures after initial construction. We used alternating least squares (ALS) to estimate a complete matrix that preserved, as accurately as possible, the covariance structure of the underlying data (*Figure 1—figure supplement 5*). To reduce run to run variation in ALS, we generated 200 complete data matrices with ALS, under different random seeds, and computed the final infilled matrix as the median of these repetitions. Simulations with ground truth data (*Figure 1—figure supplement 5*) for comparison of infilling methods were run with data randomly drawn from a multivariate normal distribution matching that of the inbred Decathlon covariance matrix. These simulated comparisons showed that median ALS imputation did not necessarily produce the infilling with the lowest mean-squared error, but did result in a correlation structure closer to ground truth than any other method. Since we are ultimately interested in the correlation structure of behavioral biases, we therefore opted for median ALS infilling. This process resulted in three complete matrices: two $n \times d$ matrices for the two Berlin-K$^{iso}$ Decathlon experiments and one $n \times d$ matrix for the NEX Decathlon experiment.

To combine the Decathlon-1 and Decathlon-2 matrices for Berlin-K$^{iso}$, we z-scored by behavioral measure within each matrix to adjust for Decathlon experiment batch effects on mean and variance. Data from the two matrices were then combined by matching unique assay and measure combination. Because the order of assays was randomized for each Decathlon experiment, day of testing was ignored when combining behavioral measures from all non-circadian assays (e.g., olfaction odorOccupancy from day 7 of Decathlon-1 was combined with olfaction odorOccupancy from day 8 of Decathlon-2). Circadian measures were matched by day of testing due to circadian measurements being collected on all days of testing (e.g., circadian meanSpeed from day 1 of Decathlon-1 was combined with circadian meanSpeed from day 1 of Decathlon-2). Placeholder NaN values were inserted in cases

where no matching measure existed in the other matrix (e.g., temporal phototaxis for Decathlon-1). The resulting full data matrix was then z-scored by measure and any residual missing values (due to lack of an existing measure match) were then infilled via the ALS method described above.

## Distilled matrix generation

We created a distilled matrix to condense any covarying features for which we had a prior expectation that the measures might be correlated for obvious or uninteresting reasons. We defined such groups of measures (a priori groups) that primarily consisted of either duplicate measures (e.g., all 10 days of circadian gravitaxis) or measures that are likely to be linked by the same underlying phenomenon (e.g., choiceNumber and meanSpeed across multiple assays). Details of the measures in each a priori group are detailed in *Supplementary file 2*. Criteria used to assign each metric to its corresponding a priori group are shown in *Supplementary file 3*. We performed PCA on each a priori group separately in an attempt to capture group variance with fewer dimensions. We defined an adaptive cutoff for the number of PCs retained from each group as the highest PC above or within the 95% confidence interval of the variance explained by PCs computed on a shuffled version of the same matrix. We reasoned that PCs above or within the variance explained for PCs computed on the shuffled matrix (a matrix with approximately independent features) represented components corresponding to meaningful covariance.

## Multidimensional analyses

To characterize the dimensional organization of the Decathlon, we computed the number of connected components in a thresholded covariance matrix representing a directionless graph. We swept 200 threshold values uniformly spaced between the minimum and maximum covariance. A spectrum of dimensional organization was formed by iteratively counting the number of connected components in the graph at each threshold value. To estimate a lower limit on the dimensionality and to assess the degree to which the number of connected components was dependent on specific groups of measures, we repeated the steps above on matrices after randomly selecting measures to drop from the matrix, iteratively increasing the number of measures dropped from one to $d$-1 (i.e., one feature remaining).

Unless otherwise specified, $t$-SNE embeddings of all data sets was performed on the Euclidean pairwise distances of $z$-scored values with perplexity = 20. All embeddings were optimized by minimizing the KL-divergence between the original and embedded data. $t$-SNE performed on Decathlon measures was run with perplexity = 8, where we had an expectation that clusters would be relatively small (e.g., 3–8 data points) due to the low number of measures in our a priori groups (median = 5) and low number of unique measures from each assay (median = 8) and day of testing.

## Unsupervised behavioral classification

As previously described (*Berman et al., 2014*), single-fly videos were decomposed into behavioral classification time series with the motion mapper pipeline. Briefly, 200 × 200 px frames centered on the flies were translationally and rotationally aligned with sub-pixel accuracy to a template fly body to restrict frame-to-frame variation to postural changes by the flies. The data dimensionality was reduced by restricting further analysis to the 6700 px with the highest variance. The data for all individuals was further compressed with PCA into 50 eigenmode postural time series. PC time series were then spectrally decomposed into 25 uniformly spaced frequency channels via Morlet wavelet transformation, resulting in a high-dimensional (1250) representation of each frame at various timescales. A representative subsampled training set of frames was constructed for each individual by embedding their data into two dimensions with $t$-SNE (*van der Maaten and Hinton, 2008*) and selecting frames according to proportionally to their local probability density in the embedded space. A joint two-dimensional embedding for all individuals data was then constructed by embedding the combined training set via $t$-SNE. Individual embeddings were generated by projecting all remaining data points into the joint embedding. As previously described, the distribution of log-speed trajectories within the embedded space was well-described by a two-component Gaussian mixture model (GMM) with the majority of frames falling into a low-speed mode. We used the standard deviation of the low-speed GMM component to define a small Gaussian kernel ($\sigma$ = 1.37) that was convolved with the embedded points the compute a continuous density. The embedded space

was then segmented into discrete regions by computing watersheds (*Najman and Schmitt, 1994*) of the negative density. Behavioral classifications were then assigned to frames by the watershed region occupied. Frames were filtered by defining an embedded speed threshold (2.78) as the intersection of GMM components where the log-speed distribution was maximally separable. All frames above speed threshold received no classification. Individual occupancy within each classification was then used to compute a discrete probability density function for each individual. The behavioral state transition matrix (*Figure 2G*) was calculated as the row-normalized probability that a fly in state $i$ (rows) would be in state $j$ (columns) 1 frame later.

Human readable labels were applied to each classification by generating 8 × 8 tiled movies of frames corresponding to individual pauses within each classification above a minimum duration of 10 ms. Labels were designated post hoc to describe the behaviors that appeared to be frequently represented after repeated viewing. The labels were composed with the following format: body part (e.g., wing, forelimb, abdomen), the behaviors displayed (e.g., walking, grooming, movement), and a qualitative descriptor of the speed of movement (e.g., idle, slow, fast).

## Statistics

Unless otherwise stated, all reported correlations were computed as the Spearman rank correlation. p-Values reported for correlation coefficients were calculated via two-tailed $t$-test of the null hypothesis that the regression coefficient was not significantly different from zero.

To generate bootstrapped distributions of correlation the correlation matrices, Decathlon matrices (either the full or distilled matrices) were sorted to match measures across Decathlon data sets as described above. A pair of bootstrapped matrices were created by bootstrapping individuals from either the same (e.g., resampling Berlin-K$^{iso}$ matrix twice) or different Decathlon matrices up to the size of the original matrix. The correlation matrix was computed for each bootstrapped matrix and the unique, off-diagonal $r$-values of each matrix were stored. A single correlation of correlations was then calculated on the two sets of $r$-values. The above steps were then repeated over 100 repetitions for all unique combinations of Decathlon data sets.

## False discovery rate calculation

We computed the FDR for significance of measure correlations as a function of $\alpha$-value for each combination of Decathlon data set (Berlin-K$^{iso}$ or NEX) and matrix type (full or distilled). Each data matrix was bootstrap resampled (100 repetitions) up to the size of the original matrix. Correlation matrices were computed for each matrix and p-values were calculated correlation coefficient (see statistics). We then calculated mean kernel density estimates for all p-values of unique measure combinations across bootstrap replicates. The above steps were then repeated for bootstrap shuffled data matrices to create a null distribution of p-values for correlation coefficients. We defined FDR as the shuffled p-value density divided by the observed p-value density. For determining significance of correlation, we set $\alpha = 0.05$, corresponding to the following FDR: Berlin-K$^{iso}$ full = 0.30, NEX full = 0.30, Berlin-K$^{iso}$ distilled = 0.37, NEX distilled = 0.42.

## Acknowledgements

We thank Ed Soucy, Brett Graham, Adam Bercu, and Joel Greenwood of Harvard's CBS Neuroengineering core for help fabricating our instruments. Julie Peng helped with the genomic experiments. Gordon Berman generously consulted on the analysis and sharing data. Timothy Sackton provided insightful guidance on the RNAseq modeling and KEGG enrichment analysis. Joshua Shaevitz kindly shared code and expertise for the unsupervised analysis imaging rig. Tanya Wolff and Gerry Rubin kindly shared most of the Gal4 lines from the circuit screen. James Crall, Jennifer Erickson, Danylo Lavrentovich, and Shradda Lall provided helpful feedback on the manuscript. ZW and KSK were supported by NSF Graduate Research Fellowships DGE-1144152 and #2013170544; JFA was supported by the NIH under grant no. GM124881. BdB was supported by a Sloan Research Fellowship, a Klingenstein-Simons Fellowship Award, a Smith Family Odyssey Award, a Harvard/MIT Basic Neuroscience Grant, the NSF under grant no. IOS-1557913, and the NIH under grant no. MH119092. We also thank the reviewers for providing thorough, helpful feedback on a hefty manuscript in the midst of a pandemic.

## Additional information

### Funding

| Funder | Grant reference number | Author |
|---|---|---|
| National Science Foundation | DGE-1144152 | Zachary Werkhoven |
| National Science Foundation | 2013170544 | Kyobi Skutt-Kakaria |
| National Institutes of Health | GM124881 | Julien Ayroles |
| Sloan Research Fellowship | | Benjamin L de Bivort |
| Klingenstein-Simons Fellowship Award | | Benjamin L de Bivort |
| Smith Family Foundation Odyssey Award | | Benjamin L de Bivort |
| Harvard/MIT Basic Neuroscience Grant | | Benjamin L de Bivort |
| National Science Foundation | IOS-1557913 | Benjamin L de Bivort |
| National Institutes of Health | MH119092 | Benjamin L de Bivort |

The funders had no role in study design, data collection and interpretation, or the decision to submit the work for publication.

### Author contributions

Zachary Werkhoven, Conceptualization, Data curation, Formal analysis, Funding acquisition, Investigation, Methodology, Software, Validation, Visualization, Writing – original draft, Writing – review and editing; Alyssa Bravin, Investigation; Kyobi Skutt-Kakaria, Pablo Reimers, Luisa F Pallares, Data curation, Investigation; Julien Ayroles, Investigation, Supervision; Benjamin L de Bivort, Conceptualization, Funding acquisition, Investigation, Methodology, Project administration, Supervision, Writing – original draft, Writing – review and editing

### Author ORCIDs

Zachary Werkhoven (iD) http://orcid.org/0000-0003-0611-5864
Luisa F Pallares (iD) http://orcid.org/0000-0001-6547-1901
Julien Ayroles (iD) http://orcid.org/0000-0001-8729-0511
Benjamin L de Bivort (iD) http://orcid.org/0000-0001-6165-7696

### Decision letter and Author response

Decision letter https://doi.org/10.7554/eLife.64988.sa1
Author response https://doi.org/10.7554/eLife.64988.sa2

## Additional files

### Supplementary files

• Supplementary file 1. Details of Decathlon behavioral assays. Assays varied in the number of metrics collected, with all assays including measurement of at least one metric repeated in another assay. In total, the assays featured four distinct arena geometries and three distinct experiment durations ranging from 15 min to 21 hr. A minimum activity criteria was applied to individuals for inclusion in the analysis from each assay, with the specific criteria depending on which metric most closely approximated activity for any particular assay.

• Supplementary file 2. Breakdown of a priori groups (group name, measures in each group, number of measures in each group, number of principal components [PCs] kept from each group, variance explained by kept PCs).

- Supplementary file 3. Decathlon assay behavioral measure details. List of all assay behavioral measure names, definitions, numeric range, calculation, assay, a priori group, and inclusion criteria used to assign behavioral measures to their a priori group. Inclusion criteria used to form a priori groups were as follows: (1) multiple measures of the same metric in the same assay across days such as circadian speed days 1–10, (2) multiple measures of the same metric across assays such as Y-maze hand clumpiness and olfaction hand clumpiness, (3) metrics that likely share a trivial coupling such as LED Y-maze bout length and LED Y-maze choice number (i.e., longer average movement bouts resulting in more traversals through the arena center), (4) measures of responses to similar stimuli such as the three phototactic measurements in the LED Y-maze, temporal shade-light, and spatial shade-light assays, and (5) a priori group formed by a single measure due to none of the previous criteria applying.

- Supplementary file 4. List of Gal4 lines grouped by the cell types they target from the thermogenetic Shibire[ts] and dTRPA1 screen. Targeted neurons varied in the number of lines tested.

- Supplementary file 5. Decathlon enriched KEGG pathways for all Decathlon experiments. Class and subclass correspond to KEGG pathway primary and secondary class terms. Observed p-value is the minimum adjusted p-value of the pathway enrichment across all behavioral metrics in the observed data (see Materials and methods details). Average minimum p-value is the mean of such p-values across bootstrap replicates. Gene hits and predictive genes correspond to the number and list of genes associated with the KEGG pathway in the observed data.

- Transparent reporting form

### Data availability

All analysis data and code generated are linked in the manuscript and supporting files at: https://zenodo.org/record/4081667#.X8U1tohKiHs Our raw tracking data from the assays covers ~600 flies over 12 days of continuous tracking between 3 and 60Hz and are greater than 50GB in size. Our unsupervised classification videos comprise ~144 million frames of uncompressed video data and are greater than 8TB in size. For these reasons, we have not posted the rawest forms of the data to online repositories, and are instead offering them upon request by external hard drive.

The following dataset was generated:

| Author(s) | Year | Dataset title | Dataset URL | Database and Identifier |
|---|---|---|---|---|
| Werkhoven Z, Bravin A, Skutt-Kakari K, Reimers P, Pallares L, Ayroles J, deBivort BL | 2020 | The Structure of Behavioral Variation Within a Genotype | https://zenodo.org/record/4081667#.X8U1tohKiHs | Zenodo, 10.5281/zenodo/779363 |

The following previously published datasets were used:

| Author(s) | Year | Dataset title | Dataset URL | Database and Identifier |
|---|---|---|---|---|
| Mackay et al. | 2020 | The *Drosophila melanogaster* Genetic Reference Panel | http://dgrp2.gnets.ncsu.edu/data.html | phenotypic data, DGRP2 |
| Robie et al. | 2017 | Mapping the Neural substrates of Behavior | http://research.janelia.org/bransonlab/FlyBowl/BABAMMATFiles/current/BehaviorData.mat | BABAM, behavioraldata |

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
