## [Decision Letter]

**Acceptance summary:**

This paper is interesting to psychologists, ethologists, and neuroscientists. It uses fruit flies as a model system and a sophisticated high-throughput experimental pipeline that captures a large space of behavioral metrics over long timescales, while keeping the identity of individual animals. Using innovative computational tools, the authors identify structure in form of covariations in this behavioral parameter space. They find that behavioral covariates appear rather sparse and high-dimensional. Hence, there are no stereotyped groups of individuality-types found in the population of flies. The work further shows that variability in behavior can be predicted in part by gene expression, and makes first attempts in targeting neuronal circuits that potentially contribute to behavioral variability.

**Decision letter after peer review:**

Thank you for submitting your article "The structure of behavioral variation within a genotype" for consideration by *eLife*. Your article has been reviewed by 3 peer reviewers, one of whom is a member of our Board of Reviewing Editors, and the evaluation has been overseen by Ronald Calabrese as the Senior Editor. The reviewers have opted to remain anonymous.

Essential revisions:

1) Please go over the reviewers detailed comments below which we believe most of them can be addressed by additional analyses, text revisions and clarifications. The authors should revise the manuscript accordingly and prepare a point-by-point response.

2) The authors should provide a more thorough analysis showing which behaviours show persistent idiosyncrasies in individual animals as opposed to trial-trial variability (see reviewer #1). Based on this, perhaps the analyses can be repeated only on those behavioural metrics.

3) One concern is that the two decathlon data are not reproducible which undermines many of the conclusions and the subsequent comparisons to outbred fly lines. See reviewer #1 (1-2) and discussion with reviewer #3. Can the authors justify better their conclusions in the face of these concerns?

4) The reviewers agree that the results in Figure 3 are very weak. Besides the finding that behavioural covariates can change with temperature in the iso line, there is little that can be learn from the circuit interrogation lines. We wonder whether the paper would be stronger without these data included.

*Reviewer #1:*

The definition of individuality and its neurogenetic basis is a fundamental problem in ethology and neuroscience. Individuals might fall into discrete groups of personality types; alternatively, individuals might be better described by a broader spectrum of independent traits. An unbiased and quantitative analysis of behavioural traits that make up an individual's personality is a prerequisite of investigating the neuronal and genetic basis of individuality. Given the technical challenges in systematically measuring many behavioural traits across sufficiently large and genetically defined populations and over long time-scales, these questions remain unanswered. This manuscript represents a tour-de-force trying to shed more light in these directions. Werkhoven and colleagues aim at characterizing structure in correlations among a large set of quantitative behavioural measures obtained from the model organism *Drosophila melanogaster*. The authors performed a large number of high throughput behavioural experiments that cover behavioural paradigms ranging from locomotion to perceptual decision-making. Data were acquired from an inbred, hence isogenic fly line, an outbred line, and various neuronal circuit manipulations. In addition, gene expression data were obtained from individuals. In this way, the authors were able to capture hundreds of behavioural metrics from hundreds of flies, while keeping their individual identities over the course of 13 days. They developed a computational analysis pipeline that quantifies the correlation matrix computed from these metrics. In a 2-step procedure, they condense this matrix into a "distilled" matrix, the entries of which contain all remaining behavioural covariates that were not a priori expected by the authors.

A central claim in this paper is that any structure in this distilled matrix should reveal the principal axes along which individuality should be described. Based on these measurements and analyses flies could not be categorized into discrete types. Moreover, behavioral covariates appear rather sparse and derive from a high-dimensional behavioral space. This would mean that each individual fly is better described by a large combinatorial set of parameters. The same qualitative finding was made between inbred and outbred flies, leading the authors to a conclusion that larger genetic diversity does not change the principal organization of behaviour. The authors perform a set of neuronal-circuit manipulations and claim in conclusion that specific neuronal activity patterns underly structure in behavioural correlations. Some correlations between gene expression and behavioral metrics were discovered, for example gene expression of metabolic pathways

can predict some variability found in the behaviour of flies. The behavioural pipeline is sophisticated and presents a great leap forward in enabling researchers to capture a large set of behavioural measures from large fly population, keeping the identity of individuals. The work is also presenting an innovative and interesting analysis pipeline.

Although we applaud these ambitious experimental paradigms and computational techniques used, we have several major reservations about this work. Reading through the manuscript multiple times, one is left confused whether the major finding is that no structure whatsoever can be found in these data and to what extent the remaining sparse correlations are of biological / ethological relevance. Another major concern arises from the high level of trial-trial variability that is found in the data, which seems to preclude identification of persistent idiosyncrasies in the behavioural traits of individuals and impedes the reproducibility of the data matrices in two repetitions of the main experiment. We feel that most of the authors' conclusions and claims are confounded by these caveats.

(1) Distinguishing persistent idiosyncrasies from trial-to-trial variability and reproducibility of decathlon data.

A major challenge in measuring personality traits or individuality is to distinguish between persistent idiosyncrasies and trial-to-trial variation; the latter could result from inherent stochastic properties of behaviors, environmental or measurement noise. To identify an idiosyncratic behavioral trait in an animal one needs to show that individuals exhibit a distinct distribution in a behavioral metric that cannot be explained by trial-to-trial variability. Such a distinction cannot be made if a behavioral metric is measured just once or during a short period, but requires repeated measures over longer time-scales from a sufficiently large population of animals. Unfortunately, in this study many measures have been taken during just one 1-2hs episode per individual of a decathlon. For other measures that were taken repeatedly (circadian assays, unsupervised video acquisition) no efforts have been undertaken by the authors to make the above distinction. Hence, the authors' conclusion that there are no "types" of flies seems premature. In Figure S1 we are surprised to see how low most behavioral measures auto-correlate when recorded on two subsequent days; most auto-correlations further drop to meaningless values when compared over time-periods that correspond to the different epochs of a decathlon. This indicates that trial-to-trial variability dominates the data. In our view it makes little sense to ask whether two behavioral metrics are correlated or not, if their autocorrelations measured over the same time-scale are already extremely low. Moreover, Figure S5B shows that the two decathlons generate largely different data matrices (correlation ~0.25), raising concerns that the results are not reproducible. We wonder whether any structure in behavioral correlations was masked by various sources of noise in this study.

Related to above, there should be error bars and number of flies for the plots in Figure S1. This figure undermines the starting point of the paper claiming persistent idiosyncratic behaviors.

(2) Given the concerns above, it is not surprising that the outbred fly line delivers another set of covariates which lack otherwise any further structure. If experiments with >100 inbred flies cannot deliver reproducible results, it cannot be expected that a similarly sized population of outbred flies would. Perhaps the needed population size must be orders of magnitudes larger in this case.

(3) Figure 3. It is intriguing to observe how the relationship between switchiness and clumpiness is perturbed upon temperature shifts. But, it seems rather uncorrelated at the restrictive temperature in the Iso line, with a slightly positive value. However, the switchiness-clumpiness correlation is not reproducible in both perturbation types at permissive temperatures. Note, that at both temperatures the Shi and Trp datasets show no – or very low correlations: the Trp lines produce correlations from approx. -0.2 (permissive T) to 0.1 (restrictive T); the Shi lines 0, 0.1 respectively. Figure 3D is very misleading in showing the best fits to the combined datasets. We are not convinced that there is a robust sign-inversion in any of these correlation. The authors' major conclusion that " thermogenetic manipulation and specific neuronal activity patterns underlie the structure of behavioral variation" is not supported by these data. The effect of temperature in the control line, although interesting, is a major caveat for interpreting the results from the Shi and Trp results.

(4) The authors measure a large set of low- and high-level behavioral metrics, e.g. walking speed and choices in Y-mazes respectively. A fundamental problem is that many of these metrics potentially have common underlying but trivial causes, e.g. covariation between speeds measured in various conditions is expected. Therefore, the authors condense their original correlation matrix (Figure 1E) into a distilled matrix (1G) by making such judgements. In the present form, it is impossible to evaluate how systematic or arbitrarily these choices were. In many cases, where the same measure was recorded repeatedly (e.g. circadian bout length) or across different conditions (e.g. mean speed) it is obvious, but for other cases it is not obvious at all for the non-expert: for example, why are circadian-bout-length and LED-Y-maze-choice-number lumped into one block of expected behavioral covariates? The current manuscript lacks detailed explanations how the authors systematically created the distilled matrix. Can the sparseness of the distilled matrix be a consequence of too generous pre-allocations? See also point (6). The bulk of the analysis in this paper is done on the "distilled matrices" which are produced by removing correlations within previously defined groups of behavioral metrics. This is said to cleanly reveal unexpected correlations, leading to a main result of the paper, the correlations between "Switchiness" and "Clumpiness". However, if the a priori categories were defined differently, then in the extreme case this correlation would have been completely removed. How sensitive is this correlation to the choice of categories, especially given that many of the Switchiness and Clumpiness metrics are from similar assays (Figure S8)?

(5) For the second pipeline that uses t-SNE and watershed (Figure 2 and S3C), a previous publication from some of the authors [1] appears to show low repeatability of this analysis. Thus, the repeatability and noise levels of the pipeline must be investigated further. These were 3x 1h recordings per decathlon. Related to comments (1-2), the authors need to show that the differences across flies (Figure 2C,D) are not expected from the level of trial-to-trial variability. Perhaps more data from individual flies need to be recorded?

(6) 1G: To our understanding, within-block entries to the distilled matrix should indicate zero correlations, because these are correlations between PCA-projections. But we see many nonzero entries. Given the information provided in the methods it is unclear why this is the case; this requires further explanation.

In any case, within-block correlations are expected to be at least very low. Hence, we expect the distilled matrix to be relatively sparse given how it was calculated. Of interest are then the across-block correlations, the authors should make this pointy more clear top the readers.

(7) Some of the author's claims are related to the spectral dimensionality description technique described in Figure S9. However, none of the real data shown in the main paper figures look qualitatively similar to the toy data. Indeed, the histograms from the main figures are on a log scale, and are thus not comparable to the toy data results. Although the technique might be well suited for certain classes of data, one interpretation of the main paper figures seems to be that no structure is revealed whatsoever. More work should be done to exclude this as a possible interpretation, at least by generating toy data that look like the real Datasets; also with respect to point (6) above.

(8) Throughout the paper, the authors use the term "independence" for orthogonal / uncorrelated datasets. Correlation/uncorrelation – dependence/independence are not interchangeable terms. To my understanding PCA decomposes into independent variables only under certain circumstances (multivariate normal distributed data). Have the authors tested for independence?

[1] Todd, J.G., Kain, J.S. and de Bivort, B.L., 2017.

Systematic exploration of unsupervised methods for mapping behavior. Physical biology, 14(1), p.015002.

The authors should be more rigorous in identifying persistent idiosyncrasies.

The authors could repeat their analysis on the behavioral parameters that show consistent auto-correlations only.

Data/results related to Figure 3 could be removed from the paper.

*Reviewer #2:*

In this paper, Werkhoven and colleagues describe a large-scale effort, using *Drosophila*, to study variation in behavior among individuals with identical genotypes, and raised in very similar environmental conditions. This addresses the important and basic question of how much behavioral variability exists under such conditions, e.g. due to stochastic processes during development. By looking across many different behaviors, the authors are able also to investigate the nature of this variability. The key conclusion of the paper is that this intragenotypic variability is high dimensional, and cannot be explained by a small set of behavioral syndromes. They find that this observation is robust to the method they use to quantify behavior, and also holds to different degrees in data sets acquired from outbred flies, or files subjected to genetic perturbations of neural activity. Furthermore, they have generated a data set that allows correlation of behavioral biases in individual animals with transcriptomic data. Altogether, this is an impressive study that, beyond its important conclusions, opens up the possibilities for many further explorations in this area, and should be interesting to a broad audience. The experiments are well designed and overall the paper is very nicely written and clear to understand.

I have a few small questions or points that were unclear to me that the authors might like to address.

1) I missed a description of the distribution across individuals of the different behavioral measures that were used for the correlation analysis. E.g. were they generally Gaussian, heavy-tailed etc?

2) One complication in their data set is that there are quite a lot of points with missing data, because a particular variable could not be measured in a particular fly. The authors deal with this by filling in the missing data using an approach which they validate using synthetic data, where they have randomly deleted a subset of points. However it appears that the missing points in their datasets are not distributed randomly (Figure S3A) but have quite a marked structure. Perhaps the authors could apply similarly structured perturbations in their toy data to strengthen the argument for the methods they use for infilling.

3) A recent preprint from the Berman lab (Hernández et al., 2020. https://arxiv.org/abs/2007.09689) ascribes some aspects of interindividual variability to temporary differences in occupancy of longer time-scale behavioral states, and they make some attempts to normalize for this in their analysis. Perhaps the authors could discuss the possible implications, if any, for interpretation of their data set.

4) In the introduction the authors cite Pantoja et al., 2016 in the context of studying intragenotypic variability, but I don't think this paper looked at individuals with identical genotypes.

5) The data presented in Figure 3 seems less convincing than the rest of the paper. First, it appears to focus on only one particular behavioral coupling so that it is difficult to make a general conclusion from this that the Decathlon experiments tend to identify behaviorally meaningful couplings. I am also confused by Figure 3D, which appears to show, in the 95% confidence interval, a very wide range of possible slopes in the control data, which overlap considerably with the result at the higher temperature. Maybe the authors can help clarify this?

*Reviewer #3:*

In this paper Werkhoven et al., ask a fundamental question in behavioral neuroscience – what is the structure of co-varying behaviors among individuals within populations. While questions in the context of inter-individual behavioral differences have been studied across organisms, this work represents a highly novel and comprehensive analysis of the behavioral structure of inter-individual variation in the fly, and the underlying biological mechanism that may shape this structure of covariation. In particular, for their experiments they combined a set of behavioral tests (some of them were explored in previous studies) to a 13-day long behavioral paradigm that tested single individuals in a highly controlled and precise way. Through clever analysis the authors interestingly showed strong correlations only between a small set of behaviors, indicating that most of the behaviors that they tested do not co-vary, exhibiting many dimensions of inter-individual variation in the data. They further used perturbations of neuronal circuits and showed that temperature and circuit perturbations can change dependencies among sets of behaviors. In a different set of experiments where they integrated gene-expression data (from the brains of single individuals), they showed that some of the genes are correlated with individual-specific parameters of behaviors. Interestingly, through comparison of inbred and outbred population they demonstrated that also outbred populations are showing relatively low covariance of behaviors across individuals.

Overall, the data in the paper indicate that surprisingly, even for a 'simple' organism, there are many dimensions of inter-individual variation, e.g. many specific characters that can change among individuals in a non-depended way. The ability of the authors to precisely measure such dependencies in such a highly robust and precise way allowed their investigation of the underlying processes that may generate this variation. The results in this study are highly interesting and novel. They uncover a general picture of the structure of behavioral variation among individuals and open many avenues for further analyses of the underlying neuronal and molecular mechanisms that control variation in sets of behaviors. Furthermore, the methods that were developed in this paper can be of great use by other researches in the field.

However, while the key claims of the manuscript are well supported by the data and analyses methods, some aspects of data analysis need to be clarified or extended.

– It is not clear what is the motivation for using the 'Effective dimensionality spectrum' analysis presented in the paper and how it significantly adds to existing methods of clustering that are relying directly on the correlation/distance matrix (some of them were used in this study).

– While it is clear that the distilled behavioral covariation matrix has many independent dimensions (as the authors indicated, most of the a-priori PCs are not strongly correlated), the number of 'significant' Pcs was not calculated directly for the distilled matrix, and t-SNE analysis is presented only for the original covariation matrix (1L).

– It is possible that some if the behaviors that covary across individuals in the high temporal resolution assay and also tend to be associated over time within an individual, may indicate sequences of behavior on longer time-scales (than the timescales in which parameters are quantified).

– Further analyses are needed for extending the detection of correlations between variation in gene-expression data and the independent behavioral measures in the covariation matrix.

The results in the study by Werkhoven et al., are highly novel and important. They provide a compelling evidence for many independent dimensions of inter-individual variation within populations. Furthermore, using their careful and accurate measurements, as well as their clever analyses, they authors interestingly demonstrated that the structure of behavioral covariation among individuals may change under specific environmental/circuit manipulations and can be predicted from gene-expression differences. For their experiments they examined ~120 different behavioral parameters (across ten behavioral paradigms) in hundreds of individuals. Their robust and precise measurements allowed them to construct a covariation matrix that represents covarying behavioral parameters across individuals, thus defining characters of individuals in a more complex way. Analyses of these data detected many dimensions of variation, uncovering a complex behavioral covariation space. They further extended their experiments to (1) perturb co-variation structure by neuronal manipulation (2) find association between gene-expression and behavioral variation (3) show that both inbred and outbred populations have many dimensions of behavioral variation (4) run their analysis pipeline on available datasets.

Overall, this is an important study that will greatly impact many areas of behavioral neuroscience. The data in the paper indicate that surprisingly, even for a 'simple' organism, there are many dimensions of inter-individual variation, e.g. many specific characters that can change among individuals in a non-dependent way. The ability of the authors to precisely measure such dependencies allowed their investigation of the underlying processes that may generate this variation. This work uncovers a more global picture of the structure of behavioral variation among individuals and open many avenues for further analyses of the underlying neuronal and molecular mechanisms that control variation in sets of behaviors. Furthermore, the methods that were developed in this paper can be of use by other researches in the field.

However, while the key claims of the manuscript are well supported by the data and analyses methods, some aspects of data analysis need to be clarified or extended.

– It is not clear what is the motivation for using the 'Effective dimensionality spectrum' analysis presented in the paper and how it significantly adds to existing methods of clustering/network analysis that are relying directly on the correlation/distance matrix (some of them were used in this study). This analysis seems more of analysis of group structure of the covariation data and not a direct dimensionality assessment such as by using PCA. Also, a possibly more direct description of X axis in Figure 1J is 'number of connected components.

– It is not clear what is the number of 'significant' Pcs in the distilled matrix (Figure 1I), using the same shuffling methods that the authors used for the original matrix. It will be interesting to include t-SNE analysis for the distilled covariation matrix (such as in 1L).

– Figure 2 – why only the second Decathlon was used (p. 6)?

– Not clear how the probability matrix in Figure 2H was generated. What are the defined/measured parameters? Not in methods.

– Temporal correlations in Figure 2H may indicate sequences of behavior on longer time-scales that can change in intensity among individuals. In this case they may covary across individuals because they are part of the same temporal sequence

– It will be interesting to correlate the 'high resolution' movement parameters (Figure 2) with the rest of the behavioral parameters (Figure 1), although there is a clear separation in timescales between behavioral parameters.

– Figure 3: "colored hash marks" – do not appear in the figure.

– Figure 4 – it will be informative to correlate gene-expression data with measures of the distilled behavioral covariation matrix (PCs), given that genes are probably correlated in the same way with behavioral parameters from the same a-priori group.

– Figure 7- Analysis of other datasets indicate that in some of the cases there are stronger correlations among behavioral parameters. It will be interesting to discuss potential sources of differences in structure between the datasets.

Figure S9 Dashed line not explained in figure caption.

Comment by reviewer #3 that arouse during the discussion, in response to reviewer #1, point (1):

The reproducibility of the two experiments- the correlations in the distilled matrix (S5B) are lower than in the original matrix (S5A). I assume that the low correlation values in the distilled matrix makes the correaltion comparison more noisy. Actually, it is hard to learn a lot by computing correlation on correlations. I would ask the authors to plot the correlations of D1, D2 (one against the other) to directly show reproducibility.

---

## [Author Response]

Reviewer #1:The definition of individuality and its neurogenetic basis is a fundamental problem in ethology and neuroscience. Individuals might fall into discrete groups of personality types; alternatively, individuals might be better described by a broader spectrum of independent traits. An unbiased and quantitative analysis of behavioural traits that make up an individual's personality is a prerequisite of investigating the neuronal and genetic basis of individuality. Given the technical challenges in systematically measuring many behavioural traits across sufficiently large and genetically defined populations and over long time-scales, these questions remain unanswered. This manuscript represents a tour-de-force trying to shed more light in these directions. Werkhoven and colleagues aim at characterizing structure in correlations among a large set of quantitative behavioural measures obtained from the model organism *Drosophila melanogaster*. The authors performed a large number of high throughput behavioural experiments that cover behavioural paradigms ranging from locomotion to perceptual decision-making. Data were acquired from an inbred, hence isogenic fly line, an outbred line, and various neuronal circuit manipulations. In addition, gene expression data were obtained from individuals. In this way, the authors were able to capture hundreds of behavioural metrics from hundreds of flies, while keeping their individual identities over the course of 13 days. They developed a computational analysis pipeline that quantifies the correlation matrix computed from these metrics. In a 2-step procedure, they condense this matrix into a "distilled" matrix, the entries of which contain all remaining behavioural covariates that were not a priori expected by the authors.A central claim in this paper is that any structure in this distilled matrix should reveal the principal axes along which individuality should be described. Based on these measurements and analyses flies could not be categorized into discrete types. Moreover, behavioral covariates appear rather sparse and derive from a high-dimensional behavioral space. This would mean that each individual fly is better described by a large combinatorial set of parameters. The same qualitative finding was made between inbred and outbred flies, leading the authors to a conclusion that larger genetic diversity does not change the principal organization of behaviour. The authors perform a set of neuronal-circuit manipulations and claim in conclusion that specific neuronal activity patterns underly structure in behavioural correlations. Some correlations between gene expression and behavioral metrics were discovered, for example gene expression of metabolic pathways can predict some variability found in the behaviour of flies. The behavioural pipeline is sophisticated and presents a great leap forward in enabling researchers to capture a large set of behavioural measures from large fly population, keeping the identity of individuals. The work is also presenting an innovative and interesting analysis pipeline.

The reviewer has identified the major contributions of the manuscript, and we are glad to hear they feel the work is innovative and interesting.

Although we applaud these ambitious experimental paradigms and computational techniques used, we have several major reservations about this work. Reading through the manuscript multiple times, one is left confused whether the major finding is that no structure whatsoever can be found in these data and to what extent the remaining sparse correlations are of biological / ethological relevance.

Technical responses to these concerns are provided below. In short, while the overall structure is one of many independent dimensions, there is structure in the form of sparse correlations between specific behaviors. Some of these correlations are not so surprising (e.g., many different measures of activity correlate across time and assay, within and between Decathlons). Some were surprising (e.g., switchiness and clumpiness) and robust in that they appear across Decathlon replicates and in wholly separate experiments. We have edited the Discussion to make it clear that these are the top-level conclusions. Specifically, we now say “We found that behavioral measures were largely independent of each other, but small sets of behaviors were significantly correlated to varying degrees. This sparse correlation structure means that with no evidence that behavioral variation cannot be readily compressed to a small number of dimensions low dimensionality.”

The pertinent sentence of the abstract has been revised to “we found sparse but significant correlations among small sets of behaviors.”

As for the ethological significance of this structure, while this is indeed an important question, particularly how this structure plays out in wild populations, directly addressing this question is beyond the scope of this lab study. We speculate that behavior having many independent axes of variation, in both inbred and outbred populations, may allow evolution to freely optimize each behavioral trait without the constraint of pleiotropic effects on other traits. In the Discussion we now say:

“The high dimensionality of the inbred and outbred matrices suggests that behaviors can evolve largely independent of one another (i.e., without pleiotropic constraint) either via plastic mechanisms within a genotype, or by natural selection in a genetically diverse population.”

It is also worth mentioning that even small but real correlations (e.g., r=0.1) can have large evolutionary effects given large populations/sufficient time.

Another major concern arises from the high level of trial-trial variability that is found in the data, which seems to preclude identification of persistent idiosyncrasies in the behavioural traits of individuals and impedes the reproducibility of the data matrices in two repetitions of the main experiment. We feel that most of the authors' conclusions and claims are confounded by these caveats.

We believe we have now improved the analysis of trial-to-trial variability and its presentation in the figures. Please see detailed replies below.

(1) Distinguishing persistent idiosyncrasies from trial-to-trial variability and reproducibility of decathlon dataA major challenge in measuring personality traits or individuality is to distinguish between persistent idiosyncrasies and trial-to-trial variation; the latter could result from inherent stochastic properties of behaviors, environmental or measurement noise. To identify an idiosyncratic behavioral trait in an animal one needs to show that individuals exhibit a distinct distribution in a behavioral metric that cannot be explained by trial-to-trial variability. Such a distinction cannot be made if a behavioral metric is measured just once or during a short period, but requires repeated measures over longer time-scales from a sufficiently large population of animals. Unfortunately, in this study many measures have been taken during just one 1-2hs episode per individual of a decathlon. For other measures that were taken repeatedly (circadian assays, unsupervised video acquisition) no efforts have been undertaken by the authors to make the above distinction. Hence, the authors' conclusion that there are no "types" of flies seems premature. In Figure S1 we are surprised to see how low most behavioral measures auto-correlate when recorded on two subsequent days; most auto-correlations further drop to meaningless values when compared over time-periods that correspond to the different epochs of a decathlon.

The reviewer is correct about the conditions that are needed to infer that a behavioral trait is idiosyncratic. We performed many experiments prior to the Decathlons to establish that most of the traits measured are idiosyncratic by this standard. In our view the autocorrelations in Figure 1—figure supplement 1 demonstrate that the behavior metrics measured include some measurement error (estimated in bootstrap CIs now reported), some trial-to-trial variability that drifts over longer timescales as well as, in many cases, biases that persist over the timescale of the decathlon experiment. This is especially evident now that Figure 1—figure supplement 1 Includes confidence intervals, see below. These experiments support the conclusion that most measures are stably idiosyncratic over the duration of the Decathlon.

We did not conduct such experiments for all behaviors analyzed in this study, as stability over days-long timescales has been established in previous studies for olfactory preferences (Honegger and Smith et al., 2020, PNAS) and behavioral modes identified by unsupervised classification (Todd et al., 2017, Physical Biology; Hernandez…Berman, 2020, BioRxiv). Additionally, at this point we have relatively strong priors that most individual behavioral measures acquired over many trials are repeatable over days as was also previously seen in locomotor handedness (Buchanan et al., 2015, PNAS), thermal and shade preference (Kain et al., 2015; Akhund-Zade et al., 2020, BioRxiv), visually guided locomotion (Linneweber et al., 2020, Science), dyadic affinity measures (Alisch et al., 2018, *eLife*) and olfactory learning (Smith et al., 2020 BioRxiv).

There is evidence of drift in biases over weeks-long timescales for most behavior measures. We interpret the fact that the autocorrelations progressively decay over multiple days of measurement suggest that this shift is biological and not due primarily to measurement error. There are indeed many behavioral autocorrelations that decay to meaninglessness within the time scale of the experiment. That is a limitation of the study, but it does not preclude us from concluding that behavior is not organized along a small number of strongly correlated axes. Nor does it preclude the possibility of measuring correlations between behavioral measures over shorter timescales. Correlations between behavioral measurements over shorter timescales could still be of ethological significance, we identified sparse significant correlations between unexpected pairs of behaviors despite the time-dependent loss of power to detect such correlations in some behaviors.

We expect that behavioral measurements are composed of many unknown variables and that many real behavioral correlations will be weak. We designed the decathlon pipeline to be high throughput so it could afford us the power to detect weak correlations. We believe that the abundance of significant day-to-day correlations that are low is indeed a key finding of our study and, in fact, what one might expect from behaviors orchestrated by overlapping circuits, but also subject to modulatory fluctuations and internal states changing on many timescales.

This indicates that trial-to-trial variability dominates the data. In our view it makes little sense to ask whether two behavioral metrics are correlated or not, if their autocorrelations measured over the same time-scale are already extremely low. Moreover, Figure S5B shows that the two decathlons generate largely different data matrices (correlation ~0.25), raising concerns that the results are not reproducible. We wonder whether any structure in behavioral correlations was masked by various sources of noise in this study.

We expected that the distribution of correlation coefficients in previous FigS5 (now Figure 1—figure supplement 6) would be quite low for the distilled matrices when comparing the first and second decathlon iterations. This is partly because our inclusion criteria only specifies that included PCs must be above or within the 95% confidence interval of the shuffled scree plot. This means we are liberally including higher PCs that may indeed reflect noisy measurements. Although the correlation of the inbred and outbred distilled matrices is relatively low overall, there are several strong relationships between distilled matrix PCs that are reproducible in inbred and outbred animals (Figure 4—figure supplement 4). These reproducible relationships suggest that the low correlation between distilled matrices is a result of (1) a biologically significant lack of correlation among the distilled behavior measures and (2) noise in some measurements, particularly the higher PCs in each a priori grouping.

We also find it encouraging that the inbred and outbred full matrices, which do not have the caveats of the a priori distillation approach, are much more correlated. This suggests that the lack of conspicuous structure common to both distilled matrices is a reflection of the lack of broad scale structure between “surprising” pairs of behaviors, rather than low reproducibility of behavioral measures.

Related to above, there should be error bars and number of flies for the plots in Figure S1. This figure undermines the starting point of the paper claiming persistent idiosyncratic behaviors.

We agree that error bars in previously Figure S1 (now Figure 1—figure supplement 1) are necessary to interpret the plot and have reorganized the figure to show the persistence of these traits in the same style as the correlation matrices reported in the main figures, as well as confidence intervals for each trait as estimated by bootstrap resampling. We believe that it can now be clearly seen that our sample sizes (in the hundreds) provided the statistical power to detect even weak correlations. Thus, our finding of few strong correlations between behaviors is a positive result, rather than reflecting a lack of power.

(2) Given the concerns above, it is not surprising that the outbred fly line delivers another set of covariates which lack otherwise any further structure. If experiments with >100 inbred flies cannot deliver reproducible results, it cannot be expected that a similarly sized population of outbred flies would. Perhaps the needed population size must be orders of magnitudes larger in this case.

Given that we had the power to detect even modest correlations between behaviors, and did detect many significant correlations in both the full and distilled matrices in inbred decathlons (Figures1E-H, 4A,G and Figure 1—figure supplement 10), we believe we had sufficient power to detect moderate changes to the correlation structure between inbred and outbred populations (particularly in the full matrix, see Figure 1—figure supplement 6, Figure 1—figure supplement 10, Figure 4—figure supplement 4).

We believe that the reviewer’s impression of a lack of shared dense structure between inbred and outbred distilled matrices is the right top-line conclusion. That said, there are specific behavioral correlations that are conserved between the inbred and outbred populations. We have added a new supplementary Figure 4—figure supplement 4 that highlights a few of these conserved relationships and also presents scatter plots of the within-decathlon correlations between the inbred and outbred experiments in the style of Figure 1—figure supplement 6.

It is certainly true that with a much larger sample size, we might have been able to (1) detect weaker correlations conserved between inbred and outbred populations, and (2) more precisely delineate the ways in which the covariation structure differs between these conditions. But we were operating very close to practical limits with this experiment, and to conduct it with substantially larger sample sizes would likely require a novel technical approach like automated fly-handling.

(3) Figure 3. It is intriguing to observe how the relationship between switchiness and clumpiness is perturbed upon temperature shifts. But, it seems rather uncorrelated at the restrictive temperature in the Iso line, with a slightly positive value. However, the switchiness-clumpiness correlation is not reproducible in both perturbation types at permissive temperatures. Note, that at both temperatures the Shi and Trp datasets show no – or very low correlations: the Trp lines produce correlations from approx. -0.2 (permissive T) to 0.1 (restrictive T); the Shi lines 0, 0.1 respectively. Figure 3D is very misleading in showing the best fits to the combined datasets. We are not convinced that there is a robust sign-inversion in any of these correlation.

We believe that at least some of the reviewer’s concerns here stem from problematic data visualization. We used a PCA-based regression to fit trend lines based on residuals that are orthogonal to the trend line. While this is appropriate given that neither of these measures stands out as a dependent vs independent variable, in combination with the z-score normalization of each measure, it produces fit lines only with slope 1 or -1. Given the modest underlying correlation, when bootstrapping is applied there is a large resulting variation in the fit lines used to determine the CI of the regression.

We are including below a comparable analysis based on standard linear regression for comparison, with switchiness as the independent variable:

This visualization of the regression supports the original interpretations. However, since this entire figure is now in the supplementary materials (Figure 3—figure supplement 1), we prefer to use the original PCA regression for consistency with the rest of the regression analyses throughout the manuscript and data browser.

The authors' major conclusion that " thermogenetic manipulation and specific neuronal activity patterns underlie the structure of behavioral variation" is not supported by these data. The effect of temperature in the control line, although interesting, is a major caveat for interpreting the results from the Shi and Trp results.

The reviewer is correct that the effect of temperature alone in the control lines limits the extent to which we can conclude that neural state affects the correlation structure. We have now re-worked the discussion of the thermogenetic experiments to emphasize the effect of temperature alone. We believe that the temperature effect on correlation is real, and quite possibly mediated through effects on brain state (c.f. work showing that latent individuality in circuit activity is revealed by thermal stress; Haddad and Marder, 2018, Neuron). However, since we did not design these experiments to test the effects of temperature, and given other reviewer’s concerns with this experiment, we have moved this figure into the supplementary materials as Figure 3—figure supplement 1.

(4) The authors measure a large set of low- and high-level behavioral metrics, e.g. walking speed and choices in Y-mazes respectively. A fundamental problem is that many of these metrics potentially have common underlying but trivial causes, e.g. covariation between speeds measured in various conditions is expected. Therefore, the authors condense their original correlation matrix (Figure 1E) into a distilled matrix (1G) by making such judgements. In the present form, it is impossible to evaluate how systematic or arbitrarily these choices were.

Although the assignment of metrics to a priori groups was a somewhat subjective process, we tried to be thoughtful in our approach. Measures grouped together met one or more of the following criteria:

1. Multiple measures of the same metric in the same assay across days (e.g. circadian speed days 1-10)

2. Multiple measures of the same metric across assays (e.g. Y-Maze hand clumpiness and Olfaction hand clumpiness)

3. Metrics that could share a plausible trivial coupling (e.g. Circadian bout length and LED Y-Maze choice number)

4. Measures of responses to similar stimuli (e.g. three measurements in response to phototactic stimuli in LED Y-Maze, Temporal Shade-light, and Spatial Shade-light)

We appreciate that this process was not transparent and have added a supplemental table to clarify our classification of each metric into its respective group. These criteria are now mentioned explicitly in Supplementary File 3.

In many cases, where the same measure was recorded repeatedly (e.g. circadian bout length) or across different conditions (e.g. mean speed) it is obvious, but for other cases it is not obvious at all for the non-expert: for example, why are circadian-bout-length and LED-Y-maze-choice-number lumped into one block of expected behavioral covariates?

As an example of the process, we reasoned that LED Y-Maze choice number and Circadian bout length would be correlated under criterion #3. Although it is not by definition true that more active animals will make more passes through the center of the Y-Maze (i.e. choices), we have observed that this is true in practice because most turn around events occur only when flies reach the end of an arm. We similarly reasoned that flies that are more active by other measures such as mean speed and choice number would also tend to move in longer bouts. Importantly, we recognized that these assumptions might be wrong, which is why we measured the compressibility of each a priori group separately and included all PCs from each group with more variance captured than shuffled controls. Indeed, there are several a priori groups that we found to be incompressible and therefore passed all the original metrics straight to the distilled matrix.

While the above explains our decision-making for this specific example, we hope the reviewer agrees that it is not essential to provide a detailed explanation for all measures in all a priori groups. We believe that the PCA process, which retains independent measures, makes our qualitative results largely robust to the subjectivity of the a priori grouping process. In addition, we now provide detailed descriptions of each measure as a supplementary table (in addition to the data browser). Along with (what we hope are) well-annotated public repositories of our data and code, it should be feasible for curious third parties to try their own alternative a priori groupings.

The current manuscript lacks detailed explanations how the authors systematically created the distilled matrix. Can the sparseness of the distilled matrix be a consequence of too generous pre-allocations? See also point (6).

Given that we grouped behaviors for which we would not be surprised to find a correlation, any additional correlations seen in a matrix constructed via less-generous grouping would, to us, be not surprising. Surprising correlations might be present in more than one “copy” in such a matrix, but the current distilled matrix should include all such redundant correlations as at least a single entry, given the within-group PCA process that retains non-compressible dimensions.

To confirm that the top qualitative conclusion of high independence with sparse significant correlations is indeed robust to the generosity of a priori group definitions, we computed a larger distilled matrix where the only a priori groups were identical measures taken across multiple days. This distilled matrix (Author response image 1) supports the same qualitative conclusions as the distilled matrix presented in the manuscript.

**Author response image 1. sa2fig1:** 

The bulk of the analysis in this paper is done on the "distilled matrices" which are produced by removing correlations within previously defined groups of behavioral metrics.

By our count, in the main figures, 37 plots present results from the full matrix and 12 plots present results from the distilled matrix. We interpret the reviewer’s comment to mean that some of the most high-level conclusions of the paper (e.g. behavioral correlations are sparse and that the structure of covariation is similar in inbred and outbred flies) were based on observations in the distilled matrix. We base our responses to the following comments on that assumption.

This is said to cleanly reveal unexpected correlations, leading to a main result of the paper, the correlations between "Switchiness" and "Clumpiness". However, if the a priori categories were defined differently, then in the extreme case this correlation would have been completely removed.

We constructed the pipeline carefully so as not to remove any correlation from the data, surprising or unsurprising. Both of these kinds of correlation are shown in the full matrix. The distilled matrix reflects our subjective definition of surprising correlations, given our experience with fly behavior.

While it is true that if we had grouped switchiness and clumpiness together in an “a priori group” (this is now in scare quotes because such a group would no longer reflect our goal of only grouping behaviors for which we had a prior expectation of correlation), then the correlation between them would have been assumed, rather than discovered. As such, the distilled matrix can be considered a conservative list of interesting correlations, with a more liberal definition of interesting being reflected in the full matrix.

How sensitive is this correlation to the choice of categories, especially given that many of the Switchiness and Clumpiness metrics are from similar assays (Figure S8)?

If we had constructed the groups differently, our quantitative interpretation of the results in previous Figure S8 (now Figure 1—figure supplement 11) would have changed, but the qualitative result would likely not have changed. If for example we did not group clumpiness measures together and switchiness measures together but grouped by assay, this correlation would have been preserved but would have been spread more broadly over different features via the loadings of the PCs that comprise the distilled matrix.

The distilled matrix shown above indicates that this less inclusive approach to defining a priori groups (a more liberal definition of what is subjectively surprising) still produces a cluster of significant correlations between switchiness and clumpiness measures.

(5) For the second pipeline that uses t-SNE and watershed (Figure 2 and S3C), a previous publication from some of the authors [1] appears to show low repeatability of this analysis.Thus, the repeatability and noise levels of the pipeline must be investigated further.

In implementing the unsupervised classification pipeline outlined in Figure 2 and previous Figure S3C (now Figure 1—figure supplement 4), we found that different runs of the embedding heuristic (the potential point of repeatability failure) gave consistent partitionings of the data. To document this, we are including a confusion matrix from two runs of the embedding algorithm in (Author response image 2).

These were 3x 1h recordings per decathlon. Related to comments (1-2), the authors need to show that the differences across flies (Figure 2C,D) are not expected from the level of trial-to-trial variability. Perhaps more data from individual flies need to be recorded?

To be clear, each fly was recorded for unsupervised analysis for only 1 h. Given the throughput of this imaging, we had to record flies over 3 separate days/batches. For this same reason, it was not feasible to include multiple recordings for unsupervised analysis per individual within the decathlon, nor over multiple consecutive days for hundreds of individuals in the style of our Figure 1—figure supplement 1 persistence experiments.

We are not certain about the reviewer’s suggestion of recording from more flies. We observed dense significant correlations among the behavioral modes Figure 2E, suggesting we had ample statistical power for detecting cross-mode correlations. While we did not conduct persistence experiments with the unsupervised classification in this study, adding more flies would not address this.

Nevertheless, we have some confidence that the measures produced by unsupervised analyses represent a similar level of day-to-day stability as the measures collected in our other assays. First, in Todd et al., we saw within-individual persistence in the distribution of behavioral modes performed, across trials separated by 24 and 48 hours (Figure 8D in that publication). The per-experiment transition matrices were similarly persistent across days within individuals (8E). Analyses from the Berman group (Berman et al., 2014, J R Soc Interface; Hernandez et al., 2020, BioRxiv) also show that differences in the profile of unsupervised behavioral modes exhibited by individuals are attributable to “long-lasting internal states'' which potentially reflect biases stable over days. Though, to our knowledge, they have not conducted test-retest experiments maintaining individual identity to distinguish hours-long from days-long internal states.

We additionally believe that the interpretation of the high-level conclusion that the correlations among behavioral motifs are organized into blocks corresponding to anatomical region is supported whether measured differences in behavioral biases reflect hours-long or days-long states.

(6) 1G: To our understanding, within-block entries to the distilled matrix should indicate zero correlations, because these are correlations between PCA-projections. But we see many nonzero entries. Given the information provided in the methods it is unclear why this is the case; this requires further explanation.

The non-zero values between PCs are the result of imperfect alignment of the covariance within inbred and outbred flies. The behavior data from both experimental groups is combined for upstream of PCA so that they share a common projection (This is necessary so that PCX of the inbred matrix can be directly compared to PCX of the outbred matrix). The data are then split back into inbred and outbred groups before performing correlation analysis. We interpret the relative low magnitude of these residual correlations as an indication that the variance is relatively well aligned across inbred and outbred flies.

In any case, within-block correlations are expected to be at least very low. Hence, we expect the distilled matrix to be relatively sparse given how it was calculated. Of interest are then the across-block correlations, the authors should make this pointy more clear top the readers.

We have added the following sentence to the caption of Figure 1G to make this clear: “Meaningful correlations in this matrix can be found outside the within-a-priori-group on-diagonal blocks.”

(7) Some of the author's claims are related to the spectral dimensionality description technique described in Figure S9. However, none of the real data shown in the main paper figures look qualitatively similar to the toy data. Indeed, the histograms from the main figures are on a log scale, and are thus not comparable to the toy data results.

We have now plotted connected components spectra in what was previously Figure S9 (now Figure 1—figure supplement 12) on a log-scale. We have done this for all such plots. We have also added analyses to Figure 1—figure supplement 12 comparing connected components spectra of the observed data to those of random data drawn sampled from the covariance matrix observed in the empirical decathlon data.

Although the technique might be well suited for certain classes of data, one interpretation of the main paper figures seems to be that no structure is revealed whatsoever.

As a philosophical point, we think it is worth respectfully remarking that correlation distributed uniformly across dimensions is indeed a structure around which behavioral variation might be organized. Not finding organization in tight dimensional bands is not a negative result or technical failure of the approach.

Insufficient motivation for the effective dimensionality spectra was brought up by reviewer 2, and we have added a couple sentences when the approach is introduced to motivate it more clearly. We now explicitly state that organization on a wide range of dimensionalities is a possible outcome: ”As an intuitive example, data filling a volume shaped like a frisbee is organized largely in two dimensions. Data shaped like a rugby ball is somewhat one-dimensional, not particularly two-dimensional and somewhat three dimensional. An approach is needed that can characterize such continuous variation in organization across dimensionalities, particularly the possibility that the data are not structured tidily in a single dimensionality.”

More work should be done to exclude this as a possible interpretation, at least by generating toy data that look like the real Datasets; also with respect to point (6) above.

We have now included in Figure 1—figure supplement 12 an example with (1) on-diagonal blocks with no correlation, (2) noise, and (3) sparse stronger correlations in off-diagonal blocks modeled from the empirical decathlon correlation matrices. This produces a dimensionality spectrum that is concordant with the observed spectrum for the distilled matrix.

(8) Throughout the paper, the authors use the term "independence" for orthogonal / uncorrelated datasets. Correlation/uncorrelation – dependence/independence are not interchangeable terms. To my understanding PCA decomposes into independent variables only under certain circumstances (multivariate normal distributed data). Have the authors tested for independence?

We thank the reviewer for catching this mistake. While we think “independent” is appropriate at several points in the paper, in the sense of linearly independent vectors or axes, data can certainly be uncorrelated without being independent. In general when we used this word in the context of PCA of real data, we meant uncorrelated. We have replaced all the misused instances of “independent” with “uncorrelated.”

[1] Todd, J.G., Kain, J.S. and de Bivort, B.L., 2017.Systematic exploration of unsupervised methods for mapping behavior. Physical biology, 14(1), p.015002.The authors should be more rigorous in identifying persistent idiosyncrasies.

We hope that with the revisions described above the statistical support for the persistence of these idiosyncrasies is now clearer.

The authors could repeat their analysis on the behavioral parameters that show consistent auto-correlations only.

We have now generated a distilled matrix using only behavioral measures with r-values greater than 0.15 for all time points in the persistence experiments (Author response image 3) . It is qualitatively similar to the original distilled matrix, suggesting that the high-level results of this approach are robust to the exclusion of less repeatable behaviors.

**Author response image 3. sa2fig3:** 

Data/results related to Figure 3 could be removed from the paper.

We are open to removing them altogether, since they are challenging to interpret. However, since they do suggest mechanistic bases for the correlation structure of behavior (physiologic broadly defined to include both temperature and neural state) we would prefer to include them in the manuscript. We believe that de-emphasizing them in the Results and Discussion, and moving Figure 3 to the supplementary materials strikes a good balance.

Reviewer #2:In this paper, Werkhoven and colleagues describe a large-scale effort, using *Drosophila*, to study variation in behavior among individuals with identical genotypes, and raised in very similar environmental conditions. This addresses the important and basic question of how much behavioral variability exists under such conditions, e.g. due to stochastic processes during development. By looking across many different behaviors, the authors are able also to investigate the nature of this variability. The key conclusion of the paper is that this intragenotypic variability is high dimensional, and cannot be explained by a small set of behavioral syndromes. They find that this observation is robust to the method they use to quantify behavior, and also holds to different degrees in data sets acquired from outbred flies, or files subjected to genetic perturbations of neural activity. Furthermore, they have generated a data set that allows correlation of behavioral biases in individual animals with transcriptomic data. Altogether, this is an impressive study that, beyond its important conclusions, opens up the possibilities for many further explorations in this area, and should be interesting to a broad audience. The experiments are well designed and overall the paper is very nicely written and clear to understand.I have a few small questions or points that were unclear to me that the authors might like to address.1) I missed a description of the distribution across individuals of the different behavioral measures that were used for the correlation analysis. E.g. were they generally Gaussian, heavy-tailed etc?

Many of the behavioral measures are roughly normally distributed, some with heavy tails. Histograms of the raw measures were previously available on the Decathlon Data Browser. We have now added them asFigure 1—figure supplement 3.

2) One complication in their data set is that there are quite a lot of points with missing data, because a particular variable could not be measured in a particular fly. The authors deal with this by filling in the missing data using an approach which they validate using synthetic data, where they have randomly deleted a subset of points. However it appears that the missing points in their datasets are not distributed randomly (Figure S3A) but have quite a marked structure. Perhaps the authors could apply similarly structured perturbations in their toy data to strengthen the argument for the methods they use for infilling.

This is a good suggestion. We have repeated the analysis in previous Figure S4 (now Figure 1—figure supplement 5) with data randomly drawn from multivariate normal distributions derived from the empirical inbred full covariance matrix. To asses the statistical impact of the empirical missing data, we took a resampling approach to generating missing values in this ground-truth sampled data. Values were deleted from the data matrix by shuffling columns of the full matrix missingness mask randomly and drawing columns up to the observed amount (40%) of missing values. We found that the infilling method performed similarly with missing data of this structure and have updated Figure 1—figure supplement 5 to show these new analyses.

3) A recent preprint from the Berman lab (Hernández et al., 2020. https://arxiv.org/abs/2007.09689) ascribes some aspects of interindividual variability to temporary differences in occupancy of longer time-scale behavioral states, and they make some attempts to normalize for this in their analysis. Perhaps the authors could discuss the possible implications, if any, for interpretation of their data set.

Our understanding of those results is as follows: the covariance structure of within-species behavior modes has similar high-level clusters as the clusters which emerge from an information bottleneck analysis that identifies the best high-level clustering of the modes such that the current mode is predictive of modes ~1h in the future. This is the longest time scale in that data because of the length of the recordings. Thus, the authors have found that the behavior is organized on hours-long timescales in ways that mirror the covariation across individuals. But this does not imply that behavior is not organized on days-long timescales in ways that also mirror the covariation across individuals. We believe the key experiment would be to test individuals on two occasions, separated by days, maintaining their identity, and our understanding is that the Berman group has not done this experiment. Given the days-long stability of unsupervised behavioral modes we observed in Todd, et al., we would not be surprised if biases in the behavioral modes identified by Berman’s approach were correlated over days.

That said, the concordance of the clustering between their within-species covariance matrix and information bottleneck analysis seems consistent with the concordance we observed between the correlation matrix in Figure 2E and the markov transition matrix between behavioral modes (2H). We now discuss this consistency in the Discussion.

4) In the introduction the authors cite Pantoja et al., 2016 in the context of studying intragenotypic variability, but I don't think this paper looked at individuals with identical genotypes.

The reviewer is correct that these animals were not deliberately inbred. However, it was a study looking at domesticated animals, which tend to have relatively low genetic diversity by virtue of their small population culture conditions. So we would guess it is pertinent to the structure of IGV. But a more relevant set of fish experiments was conducted in clonal Amazon mollies (Bierbach and Laskowski, Wolf, 2017, Nat. Comm) which we now cite instead.

5) The data presented in Figure 3 seems less convincing than the rest of the paper. First, it appears to focus on only one particular behavioral coupling so that it is difficult to make a general conclusion from this that the Decathlon experiments tend to identify behaviorally meaningful couplings.

Indeed, these focused experiments do not provide a broad validation of the many correlations observed in the Decathlon experiments. We think the experiments presented in the previous Figure 3 (now Figure 2—figure supplement 1) serve the purpose of exploring a potential mechanistic origin of the correlations between behaviors, namely physiological states.

I am also confused by Figure 3D, which appears to show, in the 95% confidence interval, a very wide range of possible slopes in the control data, which overlap considerably with the result at the higher temperature. Maybe the authors can help clarify this?

The wide range of fits seen in those confidence intervals, reflect our choice of a PCA-based regression, which minimizes residuals orthogonal to the regression line, rather than typical regressions that minimize residuals on the dependent variable (y) axis. In combination with z-score normalization applied to each behavioral measure, this approach produces regression lines that have slope 1 or -1. For a weak correlation like the one in 3D, bootstrapping produces best flt lines of both of these slopes frequently, and consequently wide CIs.

Moreover, since these experiments include changes to the correlation structure in control lines subject to the heatshock, we cannot definitively interpret the results as implying that neural state (rather than physiological state more broadly construed). For these reasons we have moved this figure into the supplementary materials, and re-emphasized the caveats associated with these experiments in the discussion.

Reviewer #3:In this paper Werkhoven et al., ask a fundamental question in behavioral neuroscience – what is the structure of co-varying behaviors among individuals within populations. While questions in the context of inter-individual behavioral differences have been studied across organisms, this work represents a highly novel and comprehensive analysis of the behavioral structure of inter-individual variation in the fly, and the underlying biological mechanism that may shape this structure of covariation. In particular, for their experiments they combined a set of behavioral tests (some of them were explored in previous studies) to a 13-day long behavioral paradigm that tested single individuals in a highly controlled and precise way. Through clever analysis the authors interestingly showed strong correlations only between a small set of behaviors, indicating that most of the behaviors that they tested do not co-vary, exhibiting many dimensions of inter-individual variation in the data. They further used perturbations of neuronal circuits and showed that temperature and circuit perturbations can change dependencies among sets of behaviors. In a different set of experiments where they integrated gene-expression data (from the brains of single individuals), they showed that some of the genes are correlated with individual-specific parameters of behaviors. Interestingly, through comparison of inbred and outbred population they demonstrated that also outbred populations are showing relatively low covariance of behaviors across individuals.Overall, the data in the paper indicate that surprisingly, even for a 'simple' organism, there are many dimensions of inter-individual variation, e.g. many specific characters that can change among individuals in a non-depended way. The ability of the authors to precisely measure such dependencies in such a highly robust and precise way allowed their investigation of the underlying processes that may generate this variation. The results in this study are highly interesting and novel. They uncover a general picture of the structure of behavioral variation among individuals and open many avenues for further analyses of the underlying neuronal and molecular mechanisms that control variation in sets of behaviors. Furthermore, the methods that were developed in this paper can be of great use by other researches in the field.However, while the key claims of the manuscript are well supported by the data and analyses methods, some aspects of data analysis need to be clarified or extended.

We are glad the reviewer saw these strengths of the paper and hope that our responses to their feedback below, and the responses to the other reviews address these aspects.

– It is not clear what is the motivation for using the 'Effective dimensionality spectrum' analysis presented in the paper and how it significantly adds to existing methods of clustering that are relying directly on the correlation/distance matrix (some of them were used in this study).

Please see detailed response below.

– While it is clear that the distilled behavioral covariation matrix has many independent dimensions (as the authors indicated, most of the a-priori PCs are not strongly correlated), the number of 'significant' Pcs was not calculated directly for the distilled matrix, and t-SNE analysis is presented only for the original covariation matrix (1L).

Please see detailed response below.

– It is possible that some if the behaviors that covary across individuals in the high temporal resolution assay and also tend to be associated over time within an individual, may indicate sequences of behavior on longer time-scales (than the timescales in which parameters are quantified).

Please see detailed response below.

– Further analyses are needed for extending the detection of correlations between variation in gene-expression data and the independent behavioral measures in the covariation matrix.

Please see detailed response below.

The results in the study by Werkhoven et al., are highly novel and important. They provide a compelling evidence for many independent dimensions of inter-individual variation within populations. Furthermore, using their careful and accurate measurements, as well as their clever analyses, they authors interestingly demonstrated that the structure of behavioral covariation among individuals may change under specific environmental/circuit manipulations and can be predicted from gene-expression differences. For their experiments they examined ~120 different behavioral parameters (across ten behavioral paradigms) in hundreds of individuals. Their robust and precise measurements allowed them to construct a covariation matrix that represents covarying behavioral parameters across individuals, thus defining characters of individuals in a more complex way. Analyses of these data detected many dimensions of variation, uncovering a complex behavioral covariation space. They further extended their experiments to (1) perturb co-variation structure by neuronal manipulation (2) find association between gene-expression and behavioral variation (3) show that both inbred and outbred populations have many dimensions of behavioral variation (4) run their analysis pipeline on available datasets.Overall, this is an important study that will greatly impact many areas of behavioral neuroscience. The data in the paper indicate that surprisingly, even for a 'simple' organism, there are many dimensions of inter-individual variation, e.g. many specific characters that can change among individuals in a non-dependent way. The ability of the authors to precisely measure such dependencies allowed their investigation of the underlying processes that may generate this variation. This work uncovers a more global picture of the structure of behavioral variation among individuals and open many avenues for further analyses of the underlying neuronal and molecular mechanisms that control variation in sets of behaviors. Furthermore, the methods that were developed in this paper can be of use by other researches in the field.However, while the key claims of the manuscript are well supported by the data and analyses methods, some aspects of data analysis need to be clarified or extended.– It is not clear what is the motivation for using the 'Effective dimensionality spectrum' analysis presented in the paper and how it significantly adds to existing methods of clustering/network analysis that are relying directly on the correlation/distance matrix (some of them were used in this study).

Our use of the effective dimensionality spectrum analysis was originally motivated by a desire to provide a spectral approach that could quantify the amount of organization at each dimension. Considering that much of the organization we observed in the behavioral data was distributed broadly over many weak correlations of varying strength, we felt that using a method that defined a single number for the dimensionality would inevitably be incomplete. We agree that the method was not sufficiently motivated in the text and that interpretation of dimensionality spectra is difficult even for the toy data sets in previous Figure S9 (now Figure 1—figure supplement 12). For that reason,(1) we have introduced the idea with new (hopefully relatable) examples, specifically considering the spectrum of dimensionality associated with data filling the volume of a rugby ball and a frisbee. (2) We have replaced the single value previously listed in the "ground truth dimensions" column of previous Figure 1—figure supplement 12 with multiple values for each example reflecting the ground-truth structure we built into these covariance matrices. This more accurately reflects the continuous distribution of organization across dimensionalities. (3) To this figure, per the suggestion of another reviewer, we have added connected components spectra based on ground truth correlation matrices constructed to match what we conclude are the key statistical features of the decathlon correlation matrices: a specific distribution of sparse correlations of varying strength. We find that spectra of this synthetic data are a qualitative match to those of the decathlon data. This suggests that these key statistical features are sufficient to capture the dimensional organization of the decathlon matrices. (4) We renamed the method to “connected components strectra” to be more transparent about the method and how it differs from other methods.

We believe these edits more clearly communicate the motivation for this analysis and the clarity of the examples in Figure 1—figure supplement 12.

This analysis seems more of analysis of group structure of the covariation data and not a direct dimensionality assessment such as by using PCA. Also, a possibly more direct description of X axis in Figure 1J is 'number of connected components.

We feel that this analysis does reflect the dimensional organization of the data, the same way that PCA (the eigen-decomposition of the covariance matrix) does. We do, however, agree that the plot labels could be clearer and have relabeled the x-axes of the dimensionality spectra “connected components” to reflect this recommendation. We have also renamed the y-axes “log(count).”

– It is not clear what is the number of 'significant' Pcs in the distilled matrix (Figure 1I), using the same shuffling methods that the authors used for the original matrix.

The scree plot of the distilled matrix is shown in Figure 1I and the number of significant PCs in the distilled matrix is reported in the text: “The full matrix contained 22 significant PCs, with PCs 1-3 explaining 9.3, 6.9 and 5.8% of the variance, respectively (the distilled matrix contained 16 significant PCs, with PCs 1-3 explaining 13.8, 10.0, and 7.7% of the variance, respectively; Figure 1I).”

It will be interesting to include t-SNE analysis for the distilled covariation matrix (such as in 1L).

The t-SNE result embedding of individual flies using the distilled matrix features is now shown in Figure 4—figure supplement 4D and shows a unimodal distribution of flies for both inbred and outbred flies to similar to flies embedding using the full matrix features. We have added a description this result to the Results section. We have not added a t-SNE embedding of the behavioral measures of the distilled matrix in the space of individual flies because 38 points is really not enough to see structure using this method.

– Figure 2 – why only the second Decathlon was used (p. 6)?

We were unable to perform the unsupervised classification analysis because the video quality of the unsupervised footage from the first decathlon was too low to reliably align frames. We improved our imaging strategy for the second (inbred batch 2) and third (outbred) decathlon iterations. The unsupervised analysis was also performed on the outbred flies but was not originally shown. We have now included Figure 4—figure supplement 2, which is comparable to Figure 2, in the supplement for the outbred flies.

– Not clear how the probability matrix in Figure 2H was generated. What are the defined/measured parameters? Not in methods.

The probability matrix in Figure 2H was generated by calculating the frame to frame probability of transitioning from state *i* to state *j* in the following frame. Therefore each row represents the transition probabilities for a given behavioral state. We have added text to the figure caption and methods to clarify this point.

– Temporal correlations in Figure 2H may indicate sequences of behavior on longer time-scales that can change in intensity among individuals. In this case they may covary across individuals because they are part of the same temporal sequence

We believe this is a very plausible scenario. It would match our observation that the clusters in the covariance matrix match those of the high-level clusters of the unsupervised embedding, as has also been reported in Hernandez et al., 2020, *BioRxiv*. The authors of that paper also note that corresponding clusters are seen in within-species covariance matrices and clusters of the embedded behavioral space defined information bottleneck criteria on hours-long timescales. While they interpret that to mean individual variation may be due primarily to hours-long states, we do not believe they can formally exclude the possibility of days-long or life-long states, as we see in the other measures of the Decathlon and in previous unsupervised clusterings of fly behavior (Todd et al., 2017, Physical Biology). Per this and another reviewer’s suggestion, we have expanded the discussion around this topic.

– It will be interesting to correlate the 'high resolution' movement parameters (Figure 2) with the rest of the behavioral parameters (Figure 1), although there is a clear separation in timescales between behavioral parameters.

We have added this as a new analysis in Figure 2—figure supplement 1. While the correlations between the unsupervised behavioral modes and the assay measures are generally weaker than the correlations within either category, there is nevertheless an enrichment for statistically significant correlations, and the structure of correlations between these two kinds of behavioral measures is roughly conserved between the inbred and outbred decathlons. This also adds confidence to our having identified consistent structure in the organization of behavior across contexts.

– Figure 3: "colored hash marks" – do not appear in the figure.

We have updated the figure (now Figure 3—figure supplement 1) caption to reflect this point.

– Figure 4 – it will be informative to correlate gene-expression data with measures of the distilled behavioral covariation matrix (PCs), given that genes are probably correlated in the same way with behavioral parameters from the same a-priori group.

The individual gene expression model p-values and enriched KEGG pathways showed some similarity across behaviors from the same a priori groups (Figure 3D, E; Figure 4—figure supplement 1 B3-4, C3-4). However, we found no significant enriched pathways for distilled matrix PCs. We now state this explicitly in the captions of Figure 4—figure supplements 1 and 3.

– Figure 7- Analysis of other datasets indicate that in some of the cases there are stronger correlations among behavioral parameters. It will be interesting to discuss potential sources of differences in structure between the datasets.

We agree that some of the data sets in this figure (now Figure 5) have stronger correlations between behavioral parameters and show much more organization at lower dimensionality. We believe that this variation in organization could reflect biological differences, as well as several non-biological causes, including:

1. Parameters calculated on simultaneous measurements – we see significantly higher correlations between measures collected on the same day. This may be in part due to shared noise. It also likely reflects increased correlation due to a broader biological state. We see that nearly all of our behavioral measurements decrease in correlation over time (see Figure 1—figure supplement 1), suggesting that only part of the behavioral variation represents a persistent idiosyncratic difference (as multiple reviewers have suggested here). Thus, data sets collected over shorter windows may exhibit higher correlations.

2. Differences in the amount of duplicated or coupled behavioral measures. JAABA collects hundreds of behavioral measures that are likely numerically or mechanically coupled (e.g. multiple measures of body angle, relative angle of several body parts between 2 flies etc). Likewise, segmentations between behavioral modes in the unsupervised pipeline are designed to be uncorrelated. In most unsupervised pipelines, the number of clusters is a “mostly free” parameter, and the more clusters are labeled, the higher the max correlation between clusters.

3. There may be differences in the signal-to-noise ratio of the measurements made across these diverse assays.

Figure S9 Dashed line not explained in figure caption.

We have improved previous Figure S9 (now Figure 1—figure supplement 12) figure in several ways: There are now several dashed lines corresponding to the multiple dimensionalities expected based on the constructed covariance matrix block-diagonal structure. They represent the peaks we would expect to see in the spectra if they can identify the dimensionalities around which the data are organized. We now say this in the caption as well.

Comment by reviewer #3 that arouse during the discussion, in response to reviewer #1, point (1):The reproducibility of the two experiments- the correlations in the distilled matrix (S5B) are lower than in the original matrix (S5A). I assume that the low correlation values in the distilled matrix makes the correaltion comparison more noisy. Actually, it is hard to learn a lot by computing correlation on correlations. I would ask the authors to plot the correlations of D1, D2 (one against the other) to directly show reproducibility.

We have now included scatter plots of the full and distilled matrix correlation values in inbred-D1, inbred-D2, outbred-D3 in Figure 4—figure supplement 4. We have also now included panels showing exemplar correlations that are conserved between inbred and outbred decathlons as pairs of scatter plots.